# Novel Soloxolone Amides as Potent Anti-Glioblastoma Candidates: Design, Synthesis, In Silico Analysis and Biological Activities In Vitro and In Vivo

**DOI:** 10.3390/ph15050603

**Published:** 2022-05-14

**Authors:** Andrey V. Markov, Anna A. Ilyina, Oksana V. Salomatina, Aleksandra V. Sen’kova, Alina A. Okhina, Artem D. Rogachev, Nariman F. Salakhutdinov, Marina A. Zenkova

**Affiliations:** 1Institute of Chemical Biology and Fundamental Medicine, Siberian Branch of the Russian Academy of Sciences, 630090 Novosibirsk, Russia; a.ilina8@g.nsu.ru (A.A.I.); ana@nioch.nsc.ru (O.V.S.); senkova_av@niboch.nsc.ru (A.V.S.); marzen@niboch.nsc.ru (M.A.Z.); 2Faculty of Natural Sciences, Novosibirsk State University, 630090 Novosibirsk, Russia; aokhina@nioch.nsc.ru (A.A.O.); rogachev@nioch.nsc.ru (A.D.R.); 3N.N. Vorozhtsov Novosibirsk Institute of Organic Chemistry, Siberian Branch of the Russian Academy of Sciences, 630090 Novosibirsk, Russia; anvar@nioch.nsc.ru

**Keywords:** soloxolone methyl, amides, 18βH-glycyrrhetinic acid, bardoxolone methyl, glioblastoma, blood–brain barrier, antitumor activity, apoptosis, mitochondrial stress, tumor microenvironment

## Abstract

The modification of natural or semisynthetic triterpenoids with amines can be explored as a promising strategy for improving their pharmacological properties. Here, we report the design and synthesis of 11 novel amide derivatives of soloxolone methyl (**SM**), a cyano enone-bearing derivative of 18βH-glycyrrhetinic acid. Analysis of their bioactivities in vitro and in silico revealed their high toxicity against a panel of tumor cells (average IC_50_^(24h)^ = 3.7 µM) and showed that the formation of amide moieties at the C-30 position of soloxolone did not enhance the cytotoxicity of derivatives toward tumor cells compared to **SM**, though it can impart an ability to pass across the blood–brain barrier. Further HPLC–MS/MS and mechanistic studies verified significant brain accumulation of hit compound **12** (soloxolone tryptamide) in a murine model and showed its high anti-glioblastoma potential. It was found that **12** induced ROS-dependent and autophagy-independent death of U87 and U118 glioblastoma cells via mitochondrial apoptosis and effectively blocked their clonogenicity, motility and capacity to form vessel-like structures. Further in vivo study demonstrated that intraperitoneal injection of **12** at a dosage of 20 mg/kg effectively inhibited the growth of U87 glioblastoma in a mouse xenograft model, reducing the proliferative potential of the tumor and leading to a depletion of collagen content and normalization of blood vessels in tumor tissue. The obtained results clearly demonstrate that **12** can be considered as a promising leading compound for drug development in glioblastoma treatment.

## 1. Introduction

To date, one of the most important semi-synthetic natural products is bardoxolone methyl (also known as RTA 402, CDDO-methyl ester, CDDO-Me) (Figure 1), which displays a wide spectrum of bioactivities, including marked antitumor, anti-inflammatory and cytoprotective effects in both cellular and animal models [1] and has reached Phase III clinical trials for chronic kidney disease in diabetic patients [2], pulmonary hypertension [3] and Alport syndrome [4]. The enhanced biological activity observed for this derivative has been attributed to a 2-cyano-3-oxo-1(2)-ene moiety [1], which has led to the creation of first-generation semi-synthetic derivatives bearing this pharmacophore based on various triterpene scaffolds, including oleanan, ursane, lupan, lanostane, etc. [5,6].

To improve the pharmacological properties of these derivatives, several research groups have reported the synthesis of the second-generation members by modifying the carboxylic group [6,7,8,9,10,11], among which the most promising are derivatives containing various functional groups linked through amide bonds [10,11,12,13]. For instance, omaveloxolone (RTA 408) (Figure 1) was found to display a similar in vitro bioactivity profile to bardoxolone methyl [14] and is currently under clinical investigation by Reata Pharmaceuticals (USA) for various indications, including Friedreich’s ataxia [15], mitochondrial myopathies [16], immuno-oncology and prevention of corneal endothelial cell loss following cataract surgery [17]. Moreover, it was shown that omaveloxolone and other amides of bardoxolone, such as CDDO ethylamide (CDDO-Et) and CDDO 2,2,2-trifluoroethylamide (CDDO-TFEA) (Figure 1), can effectively pass the blood–brain barrier (BBB) in animal models [11,18] and ameliorate various brain disorders, including ischemic optic neuropathy [19], amyotrophic lateral sclerosis [11] and subarachnoid hemorrhage-induced brain injury [20]. Interestingly, despite the proven BBB permeability of these compounds and their marked antitumor activity in vitro [21,22,23] and in vivo [23], the studies evaluating their inhibitory effects on the growth of brain tumor cells have scarcely been reported. To the best of our knowledge, the attention of researchers in this field has mainly been focused on the investigation of pro-apoptotic and cell cycle-arresting properties of the compounds [22,24], whereas the evaluations of their anti-metastatic potential (clonogenicity, motility, invasiveness) in vitro and their antitumor activity in animal models of glioblastoma and neuroblastoma have not yet been published.

There are many examples that the modification of triterpene compounds, including bardoxolone, with various amines leads to semisynthetic derivatives with improved properties compared to the starting materials [25,26,27,28]. Recently, Huang et al. synthesized a series of cytotoxic ursolic acid amides by reacting the carboxyl group at the C-28 position with various amines, among which the compound functionalized by *N*^1^,*N*^1^-dimethylethane-1,2-diamine moiety was found to be the most potent due to enhanced cytotoxicity in a panel of malignant cells, its ability to trigger apoptosis and cell cycle arrest as well as to inhibit NF-κB signaling in human NCI-H460 lung cancer cells [25]. Furthermore, modifying the carboxyl group at C-28 of 23-O-acetyl-3-oxo-23-hydroxybetulinic acid with ethylenediamine significantly increased its antitumor activity not only against various tumor cell lines in vitro, but also against the growth of murine H22 hepatoma and B16 melanoma in vivo [26]. Additionally, amides of 18βH-glycyrrhetinic acid, containing both an indole fragment and an amine fragment in the side chain, exhibited high antitumor activity against a large panel of tumor cell lines of different histological origins, including prostate, breast and non-small cell lung adenocarcinoma, glioblastoma and melanoma [27].

Previously, our group synthesized and characterized soloxolone methyl (SM), a first-generation cyano enone-bearing derivative of 18βH-glycyrrhetinic acid [29]—a structural isomer of bardoxolone methyl at the ester location (Figure 1). We found that SM induced tumor cell death via mitochondrial apoptosis [30] and endoplasmic reticulum stress [31,32], markedly suppressed the epithelial–mesenchymal transition of tumor cells [33] and effectively inhibited the growth of murine Krebs-2 carcinoma [30] and metastasis of murine B16 melanoma [33] in vivo.

Here, we report the design, synthesis and biological studies of novel soloxolone amides containing promising pharmacophore groups. These members of the second generation of cyano enone-bearing triterpenoids were synthesized using aliphatic and aromatic amines containing additional functional groups (hydroxy, methoxy, *N*,*N*-dimethyamine, bis-*tert*-butylphenol indole, *para*-substituted anilines and 3-aminopyridine) (Figure 1). The screening of a spectrum of biological activities of novel derivatives was performed, followed by their structure–activity relationship (SAR) analysis. Given the previously demonstrated ability of bardoxolone amides to pass the BBB, the BBB permeability of novel soloxolone amides was predicted in silico and verified in a murine model in vivo. The cytotoxicity of the revealed BBB-permeable derivatives against glioblastoma and neuroblastoma cells was further explored. Thereafter, cell death-induced mechanisms of hit compound **12**, soloxolone tryptamide, as well as its antitumor potential in non-toxic concentrations were evaluated in glioblastoma cells. Finally, high anti-glioblastoma activity of compound **12** was verified in a U87 glioblastoma xenograft model in vivo.

## 2. Results and Discussion

### 2.1. Chemistry

We obtained a set of amides containing various aliphatic and aromatic substituents in the amine moiety using **SM** as the starting material for the synthesis (Figure 1).

First, we transformed the methyl ester into a carboxylic group. The general method for the conversion of methyl ester to a carboxylic group, including those of triterpene nature, is halogen-mediated hydrolysis with lithium iodide in *N*,*N*-dimethylformamide (DMF) under reflux [7,34,35]. A significant disadvantage of this method is the high reaction temperature which decreases the yield of the product: using this approach, soloxolone (**1**) was obtained with a 30% yield [36]. Based on the fact that the esterification is a reversible reaction, we hydrolyzed the ester group using inorganic bases in aqueous–alcoholic media. The best yield of compound **1** was obtained when the reaction was carried out in lithium hydroxide in aqueous ethanol at 65 °C. Under these conditions, soloxolone (**1**) was obtained with 60% yield after purification by column chromatography.

Amides (**2**–**12**) were synthesized through peptidic coupling reactions between soloxolone (**1**) and various primary amines (Figure 1). These reactions were performed using pre-activation of carboxylic groups with *N*,*N*′-carbonyldiimidazole (CDI) and desired derivatives were obtained with 30–80% yields after purification by flash chromatography.

The successful preparation of the derivatives **2**–**12** was confirmed by the presence of a δ signal corresponding to the amide proton (C(O)N*H*) in the ^1^H nuclear magnetic resonance (NMR) spectrum: at ~6.6–7.0 ppm as a broad triplet for compounds (**2**–**7**, **11**–**12**) and at ~8.4–8.6 ppm as a broad singlet for compounds (**8**–**10**). On the ^13^C NMR spectrum, the presence of the amide bond was confirmed by the presence of a δ signal at ~175–176 ppm. Furthermore, the additional groups in the side chain could be detected by the presence of extra δ signals in ^1^H and ^13^C NMR spectra, varying in accordance with the functional groups.

### 2.2. Biological Evaluation

#### 2.2.1. Cytotoxicity of Novel Compounds

In the first step of the study, we analyzed the cytotoxicity of novel soloxolone derivatives **1**–**12** in a panel of tumor cells, including human cervical carcinoma HeLa, human duodenal carcinoma HuTu-80 and murine melanoma B16 cells, as well as human non-transformed hFF3 foreskin fibroblasts. The cells were treated with derivatives for 24 h and cell viability was evaluated by 3-(4,5-dimethylthiazol-2-yl)-2,5-diphenyltetrazolium bromide (MTT) assay. **SM** was used as the reference compound. The obtained IC_50_ values of novel derivatives were summarized in Table 1.

As is evident from Table 1, the hydrolysis of the ester group of **SM** led to a significant reduction in cytotoxicity: soloxolone (**1**) was found to display an average IC_50_ value of 8.2 µM versus 2.1 µM for **SM** (Table 1). The soloxolone amides (**2**–**12**) demonstrated high antitumor activity at micromolar concentrations and their IC_50_ values were mostly similar to those of **SM** (Table 1). Comprehensive analysis of the cytotoxic profiles of novel compounds revealed some structure–activity relationships:The presence of an additional *N*,*N*-dimethylamine-containing group in the side chain of soloxolone amides led to a decrease in the cytotoxicity of the derivatives compared to **SM** in a panel of tumor cells (average IC_50_**^6^** = 3.0 µM and IC_50_**^7^** = 4.5 µM versus IC_50_**^SM^** = 2.1 µM).Hydroxyamides with C-2 and C-5 linkers (compounds **2** and **4**, respectively) had similar cytotoxic activity compared to **SM** (average IC_50_**^2^** = 1.9 µM and IC_50_**^4^** = 2.2 µM versus IC_50_**^SM^** = 2.1 µM in a panel of tumor cells); the cytotoxicity of hydroxyamide with a C-3 linker was somewhat lower (average IC_50_**^3^** = 3.4 µM).The introduction of aryl-containing moieties into the side chain of soloxolone amides had virtually no effect on the toxicity of derivatives against tumor cells (average IC_50_**^8^**^−**11**^ = 1.5–1.8 µM versus IC_50_**^SM^** = 1.5 µM); however, it slightly increased their cytotoxicity against fibroblasts (average IC_50_**^8,10−12^**^(hFF3)^ = 1.5 µM versus IC_50_**^SM^** ^(hFF3)^ = 4 µM);In the case of hFF3 fibroblasts, compounds **3** and **7** with a C-3 linker between functional groups, both in the series of hydroxyamides (compounds **2**–**4**) and dimethylaminoamides (compounds **6**–**7**), respectively, had a somewhat lower toxicity compared to their C-2/C-5 linker-containing counterparts (IC_50_**^3^** = 6.8 µM versus IC_50_**^2^** = 1.2 µM (C-2 linker) or IC_50_**^4^** = 3.6 µM (C-5 linker); IC_50_**^7^** = 7.7 µM versus IC_50_**^6^**= 4.1 µM (C-2 linker)). Additionally, the replacement of hydroxy or dimethylamino groups with a methoxy moiety was desirable for soloxolone amides with a C-2 linker (IC_50_**^5^** = 12.8 µM versus IC_50_**^2^** = 1.2 µM and IC_50_**^6^** = 4.1 µM).

Hierarchical clusterization of the cytotoxic profiles of the investigated derivatives revealed four main clusters: (1) compounds **4**, **6** and **9** showed similar cytotoxic signatures to **SM**; (2) aryl-containing compounds **2**, **8** and **10**–**12** displayed the highest overall cytotoxicity; (3) compounds **3** and **7** were more toxic for HuTu-80 cells; and (4) compounds **1** and **5** were characterized by the highest selectivity for tumor cells (Figure 2). The obtained results showed that the overall cytotoxicity of the evaluated compounds was reduced in the order: **12** > **8** ≈ **10** ≈ **11** > **2** > **SM** > **4** ≈ **9** > **6** > **3** > **7** ≈ **5** > **1**. HuTu-80 cells were found to be more susceptible to synthesized derivatives compared with other evaluated cell lines (average IC_50_^HuTu-80^ = 1.7 µM versus IC_50_^HeLa,B16,hFF3^ = 2.2–5.7 µM) (Table 1).

Thus, the cytotoxicity profiling of novel compounds clearly demonstrates that their toxic characteristics mainly depend on a cyano enone-bearing triterpenoid scaffold and amide-containing moieties only slightly modulate this activity. However, the introduction of such groups can markedly change the physicochemical parameters of the derivatives and thus can improve their pharmacokinetic characteristics—something that will require further research. The similar cytotoxicity rate of novel derivatives of **SM** in both malignant cells and non-transformed fibroblasts that was revealed can be explained by the different tissue sources of the explored cells. To elucidate the tumor selectivity of soloxolone amides more comprehensively, evaluation of their toxic profiles in tumor-bearing mice is required. Considering the proven ability of amide derivatives of bardoxolone methyl to pass across the blood–brain barrier (BBB) [10,18], our attention further turned to the study of BBB permeability of synthesized compounds and their antitumor potential against brain tumor cells in silico, in vitro and in vivo.

#### 2.2.2. Analysis of the BBB Permeability of the Novel Compounds In Silico

Given the known ability of amide derivatives of bardoxolone methyl to be successfully accumulated in the brain tissues of mice [10,11] and cynomolgus monkeys [18], next we questioned whether the synthesized amides of soloxolone are able to penetrate the BBB. For this, their BBB permeability was predicted in silico using two independent chemoinformatics tools, AlzPlatform and PreADMET, based on different algorithms of quantitative structure–activity relationship (QSAR) descriptor analysis, including SVM/LiCABEDs (machine learning) [38] and resilient back-propagation (neural network analysis) [39], respectively. As depicted in Figure 3A, SM cannot effectively pass the BBB, which agrees well with the previously published data. Sporn et al. [40] showed a very low brain uptake capacity of bardoxolone methyl in a murine model. This fact, along with a sufficient BBB score for temozolomide, a first-line agent for glioblastoma treatment, as well as an inability of doxorubicin to penetrate the BBB according to both predictive tools (Figure 3A), clearly demonstrates the good reliability of chemoinformatics data. It was found that most analyzed soloxolone amides can cross the BBB, among which soloxolone tryptamide **12** was characterized by the highest BBB score values computed by both algorithms. The overlapping of the obtained lists of BBB-permeable derivatives identified eight common compounds (**4**–**10**, **12**) which were selected for further study (Figure 3B). Besides these, derivative **11** was also included in the final list of compounds of interest because of its outstanding BBB score as calculated by the PreADMET tool (Figure 3A).

It is known that P-glycoprotein (P-gp) localized at the luminal membrane of endothelial cells plays a key chemical barrier function in the BBB, mediating the efflux of xenobiotics from the brain to the blood [41]. To evaluate whether selected derivatives are transported by P-gp, their P-gp substrate specificity was predicted by three independent web-tools, including ADMETlab [42], SwissADME [43] and LiverTox [44]. The obtained data demonstrated the low susceptibility of all analyzed compounds to P-gp transport, among which *p*-substituted anilides (**8**, **9**) and tryptamide **12** were predicted as non-substrates of P-gp by all the chemoinformatics tools used (Figure 3C, left heatmap). The computed results are in line with published data: doxorubicin is a known substrate of P-gp [45], whereas **SM** was found to exhibit similar cytotoxicity against cervical carcinoma cells independently on their P-gp expression status [29,31]. Moreover, in silico analysis also revealed that the evaluated molecules can inhibit the efflux activity of P-gp (Figure 3C, right heatmap). Considering that P-gp efflux pump inhibition is a promising approach to increase the transport of brain-targeted drugs through the BBB [46], a combination of investigated triterpenoids with brain-targeted therapeutics can significantly increase the efficiency of the latter.

Altogether, the predicted BBB permeability of novel soloxolone amides along with the absence of their P-gp substrate specificity can mediate their efficient accumulation in brain tissue, which encourages their further investigation as therapeutic candidates for treating brain disorders. Since the obtained results were predicted by chemoinformatics tools, their further experimental validation is required.

#### 2.2.3. Verification of BBB Permeability of Soloxolone Tryptamide (12) In Vivo

In the next step of the study, the ability of synthesized compounds to pass the BBB was verified in a murine model. Given that soloxolone tryptamide (**12**) displayed one of the highest BBB scores (Figure 3A) and the absence of P-gp substrate specificity according to several chemoinformatic tools (Figure 3C), this derivative was chosen as a hit compound to validate our in silico data.

Compound **12** was intraperitoneally injected at a dosage of 50 mg/kg in nude mice thrice a week for two weeks (seven injections were performed in total) and brain tissue was collected 48 h after the last injection of **12** followed by a high-performance liquid chromatography–tandem mass spectrometry (HPLC–MS/MS) analysis of the obtained samples (Figure 3D). This treatment regimen was chosen due to its common usage in the evaluation of antitumor activity of various low-molecular weight compounds in xenograft murine models [47,48,49]. Analysis of mouse body weight monitored during the experiment did not reveal significant changes compared to untreated mice, which demonstrated the absence of systemic toxicity (data not shown). Before the HPLC–MS/MS analysis, chromatographic conditions and mass spectrometer parameters for detecting both semisynthetic triterpenoid **12** and the internal standard in the multiple reaction monitoring (MRM) mode were optimized. The optimized parameters are given in Appendix A.

To estimate the content of **12** in mouse brain tissue, a set of calibrators was prepared followed by their preparation protocol. The obtained samples were analyzed using the developed HPLC–MS/MS method and a calibration curve was built over the range of concentrations of 10–1000 ng/g. We should note here that this study aimed to estimate the content of **12** in mouse brain tissue only; to obtain more reliable data, the method used has to be further validated in accordance with corresponding regulatory documents [50,51]. The validation of the method for quantification of **12** in mouse brain tissue and investigation of its pharmacokinetics will be the objectives of our next study.

The analysis of brain tissue samples collected from the animals 48 h after the last administration of the compound at a dosage of 50 mg/kg revealed **12** in all samples at concentrations of 490 ng/g (Figure 3E). Thus, our data clearly demonstrate the ability of soloxolone tryptamide **12** to successfully pass the BBB, confirming our in silico data (Figure 3A) and showing the advantageousness of further study of the antitumor potential of novel amide derivatives of soloxolone in brain tumor cells.

#### 2.2.4. Cytotoxicity of Novel Soloxolone Amides (4–12) against Brain Tumor Cells

To evaluate the antitumor activity of the synthesized soloxolone amides in brain tumor cells, the cytotoxicity of compounds **4**–**12**, predicted as BBB-permeable (Figure 3), was further analyzed in human glioblastoma U87 and U118 cells, as well as human neuroblastoma KELLY and murine neuroblastoma Neuro2a cells. As in the case of non-brain tumor cells (Table 1), the obtained results demonstrated that the novel compounds displayed similar bioactivities to reference compound **SM** (IC_50_**^4^**^−**12** (median)^ = 1.9 µM; IC_50_**^SM^**
^(median)^ = 1.4 µM) (Table 2). It was found that only dimethylaminoamides (**6**–**7**) and methoxyamide (**5**) of soloxolone showed slightly lower cytotoxicity (IC_50_**^5^**^–**7** (average)^ = 3.8 µM) compared with other derivatives (IC_50_**^4^**^,**8**–**12** (average)^ = 1.8 µM) (Table 2). 

The susceptibility of brain tumor cells to the evaluated semisynthetic triterpenoids was similar to that of non-brain tumor cells (Table 1) and decreased in the order of KELLY > Neuro2a > U87 > U118. The high toxicity of the synthesized compounds at low micromolar concentrations identified in brain malignant cells, along with their predicted ability to pass the BBB, clearly shows their suitability as a platform to develop novel anti-glioblastoma and anti-neuroblastoma agents. To understand how particularly novel semisynthetic triterpenoids induce tumor cell death, soloxolone tryptamide **12** with proven ability to pass the BBB (Figure 3E) and promising toxicity against brain malignant cells (IC_50_ = 1.6–1.9 µM) was chosen for further mechanistic studies. Given that glioblastoma is the most aggressive and lethal type of brain malignant disorders [52] and glioblastoma U87 and U118 cells are characterized by high susceptibility to compound **12** (Table 2), the mentioned cells were used as experimental models in our further work. Both cell lines belong to glioblastoma multiforme cells and have high proliferation and metastatic rates [53]. Despite the similar proliferative characteristics of U87 and U118 cells, U87 cells are known to be more aggressive compared with U118 cells, exhibiting a more invasive phenotype [54] and more efficient formation of spheroid structures [55].

#### 2.2.5. Soloxolone Tryptamide **12** Induced Apoptosis-Dependent and Autophagy-Independent Death of Glioblastoma Cells

To evaluate which form of cell death was induced by compound **12** in glioblastoma cells, its ability to stimulate apoptosis, necrosis or autophagy was examined. Double staining of compound **12**-treated U87 and U118 glioblastoma cells with annexin V-APC and propidium iodide (PI) clearly showed that **12** triggered cell death by the apoptotic pathway (Figure 4A,B). The treatment of U87 cells with **12** for 24 h induced accumulation of late apoptotic cells in a dose-dependent manner from 5.9% in the control to 28% and 73.4% in cells incubated with 2 µM and 4 µM of **12**, respectively (Figure 4A, right upper quadrant) and did not cause necrotic cell death (Figure 4A, left upper quadrant).

Since **12** at 2 and 4 µM significantly induced apoptosis in U87 cells (Figure 4C), these concentrations of **12** were further examined in U118 cells. Similar to U87 cells, **12** was found to trigger apoptosis in U118 cells in dose-dependent manner (Figure 4B,C). Moreover, the observed effect was also time-dependent: the treatment of U118 cells with **12** at 4 µM for 6, 18 and 24 h induced apoptosis in 11.7%, 34.4% and 41.5% of cells, respectively (Figure 4B). Considering that the number of necrotic cells in different samples did not exceed 10% (Figure 4B), it can be concluded that **12** predominantly causes apoptotic death of glioblastoma cells.

To double check the pro-apoptotic activity of **12**, its effect on the activation of executioner caspase-3 and -7, key markers of apoptosis, was evaluated in glioblastoma U87 cells. Using CellEvent^TM^ Caspase-3/-7 reagent, it was found that the treatment of U87 cells with **12** effectively activated caspase-3 and -7 in a time-dependent manner (Figure 4D) which independently confirmed the pro-apoptotic effect of **12** in glioblastoma cells.

Given that pentacyclic triterpenoids can induce tumor cell death not only by the apoptotic pathway but also by triggering autophagy [56], we next studied whether **12** stimulated autophagy in glioblastoma cells. For this, U87 and U118 cells were incubated with tryptamide **12** at toxic concentrations (2 or 4 µM) for 24 h followed by cell staining with monodansylcadaverine (MDC), a fluorescent dye specifically labeling autophagosomes [57]. As shown in Figure 5A, incubation of cells with **12** led to significant accumulation of MDC-positive cells, and U87 cells were more susceptible to this effect compared with U118 cells. Considering that autophagy can be associated not only with cell death but also with cytoprotection [58], its role in the cytotoxicity of **12** in glioblastoma cells was further studied. Using chloroquine, a well-known autophagy inhibitor, it was found that autophagy induced by **12** in U87 cells was not associated with cell death: an incubation of tryptamide **12**-treated U87 cells with chloroquine did not affect the cytotoxic activity of the investigated triterpenoid (Figure 5B).

Thus, the mechanistic studies performed clearly demonstrated that soloxolone tryptamide **12** induced death of glioblastoma cells by the activation of caspase-mediated apoptosis and in an autophagy-independent manner.

#### 2.2.6. Soloxolone Tryptamide **12** Induced ROS-Dependent Death of Glioblastoma Cells by Disruption of Mitochondrial Homeostasis

Given the facts that mitochondria play a key role in the regulation of cell death [59] and that pentacyclic triterpenoids are known mitochondrial stressors [56], we sought to determine whether the pro-apoptotic effect of **12** in glioblastoma cells includes the perturbation of mitochondrial homeostasis. To check this, the influence of **12** on mitochondrial polarization, mitogenesis and reactive oxygen species (ROS) production in U87 and U118 cells was evaluated.

It is known that structural analogs of compound **12**, such as SM, bardoxolone methyl and CDDO-Im, trigger the mitochondrial pathway of apoptosis in tumor cells, leading to significant dissipation of mitochondrial membrane potential (Δψ_M_) [56]. To evaluate the ability of **12** to affect Δψ_M_ in glioblastoma cells, U118 cells were incubated with **12** (4 µM) for 6 and 24 h and stained with JC-1, a fluorescent probe accumulated either within highly polarized mitochondria emitting red fluorescence (JC-1 aggregates) or in cytoplasm, in the case of low Δψ_M_ (JC-1 monomers, green fluorescence). The obtained results showed that **12** increased the percentage of U118 cells with collapsed Δψ_M_ up to 7.5% and 20.4% at 6 and 24 h, respectively versus 4.7% in control cells (Figure 6A) and significantly reduced JC-1 aggregate/monomer ratios in the cells (Figure 6B).

In order to doublecheck the Δψ_M_-targeted effect of **12** and to assess the correlation between dissipation of Δψ_M_ and induction of apoptosis in triterpenoid-treated glioblastoma cells, U87 cells were incubated with **12** at 4 µM for 6–24 h followed by their staining with a mix of MitoTracker Red, a fluorescent dye accumulated within mitochondria depending on their Δψ_M_, and fluorophore-conjugated annexin V, used to visualize apoptotic cells. As shown in Figure 6C, the incubation of U87 cells with **12** for 6, 18 and 24 h increased the percentage of apoptotic cells with dissipated Δψ_M_ from 5.7% in the control to 12.2%, 18.7% and 28% in experimental samples, respectively, and the observed effect was statistically significant (Figure 6D). Taken together, these results clearly indicate that the pro-apoptotic activity of **12** is mediated by the disruption of mitochondrial functions in glioblastoma cells.

Next, to evaluate whether the observed compound **12**-driven disturbance of mitochondrial homeostasis in glioblastoma cells involved an alteration in mitochondrial abundance, U87 cells were treated with **12** at 2 and 4 µM for 24 h and stained with MitoTracker Green, a fluorescent probe accumulated in mitochondria Δψ_M_-independently. The flow cytometry analysis demonstrated that **12** at 4 µM increased mitochondrial mass 1.6-fold in U87 cells compared to the control (Figure 6E,F), which can be explained by the induction of mitogenesis in tumor cells in response to **12**. A lower concentration of **12** (2 µM) was also found to induce mitochondrial biogenesis in glioblastoma cells; however, this alteration was statistically insignificant (Figure 6E,F). The obtained results agree well with published data: Yang et al. [60] and Gibellini et al. [61] found a similar increase in mitochondrial mass in RAW264.7 macrophages and colon carcinoma RKO cells treated with CDDO-Im and CDDO, respectively. We suppose that the observed triterpenoid-induced mitogenesis in glioblastoma cells is mediated by compensatory mechanisms in response to compound **12**-triggered mitochondrial stress and that it plays a cytoprotective role—high mitochondrial mass is known to be highly correlated with chemotherapy-driven redundant production of reactive oxygen species (ROS) in tumor cells [62] and to underlie their chemoresistance [63].

To check the ability of **12** to influence ROS production, U87 cells treated with **12** (4 µM) for 24 h were stained with dichlorodihydrofluorescein diacetate (DCFDA) and analyzed by flow cytometry. It was found that **12** increased the generation of ROS 1.9-fold in U87 cells compared to the control (Figure 6G,H). These observations are in line with the revealed mitochondria-impairing effects of **12** and published data. On the one hand, the disruption of mitochondrial homeostasis by phytochemicals was found to cause excessive ROS generation and oxidative stress induction in tumor cells [64]; on the other hand, aberrant ROS formation can lead to the dissipation of Δψ_M_ [65]. Since ROS play an important role in the regulation of cell death [64], we further examined whether ROS production mediated the cytotoxicity of **12** with respect to glioblastoma cells. The addition of N-acetyl-L-cysteine (NAC), a known free radical scavenger, to U87 cells treated with **12** resulted in complete abrogation of the cytotoxic effect of the investigated triterpenoid (Figure 6I), which clearly demonstrates ROS-dependent induction of glioblastoma cell death by soloxolone tryptamide **12**.

Previously, bardoxolone (CDDO) was found to directly interact with mitochondrial LonP1 protease and effectively inhibit its peptidase activity [66]. LonP1 is a key regulator of mitochondrial homeostasis, mediating the selective degradation of mutant and abnormal proteins in mitochondria [67]. Numerous studies have reported that silencing or pharmacological inhibition of LonP1 in human tumor cells or fibroblasts results in significant dissipation of Δψ_M_ [68,69,70] and aberrant ROS generation [68,71]. Given these facts, along with the structural similarity of **12** and bardoxolone, we supposed that the observed stressor effects of **12** on mitochondria can be explained by its direct interaction with LonP1. To prove this hypothesis, a molecular docking analysis of **12** was simulated with the crystal structure of mitochondrial LonP1 (Protein Data Bank (PDB) ID: 6X27) (Figure 6G). The top-scoring binding pose of **12** was characterized by low binding energy (ΔG = −8.8 kcal/mol) and the formation of strong hydrogen bonds with Ser855 and Lys898 which formed the catalytic dyad of LonP1 [72]. Additionally, the triterpenoid core of **12** was stabilized by a hydrogen bond with Asp852 (Figure 6G), playing an important role in the enzymatic activity of LonP1 [73].

Taken together, our results demonstrate that the revealed pro-apoptotic activity of compound **12** in glioblastoma cells can be mediated by its direct inhibition of LonP1, leading to massive mitochondrial stress, resulting in excessive ROS production and depolarization of mitochondria. Observed mitogenesis in response to **12** can be explained by the attempts of glioblastoma cells to replenish the energy supply of the cells disturbed by the investigated compound; however, this cytoprotective mechanism was insufficient to effectively prevent the mitochondria-impairing effect of **12**.

#### 2.2.7. Soloxolone Tryptamide **12** Displayed Significant Antitumor Potential against Glioblastoma Cells at Non-Toxic Concentrations

To evaluate the antitumor potential of **12** more comprehensively, its effects at non-toxic concentrations (25–500 nM; cell survival of ≥90%) on a range of metastatic-related processes in glioblastoma cells were studied. Firstly, we investigated the inhibitory activity of **12** on the clonogenic capacity of tumor cells in vitro, which reflects their ability to form colonies from single cells which characterizes the stemness of tumor cells and plays a crucial role in metastasis [74]. It was found that treatment of U87 and U118 cells seeded at low density with **12** resulted in marked suppression of the clonogenic activity of glioblastoma cells (Figure 7A). U87 cells were shown to be more susceptible to the investigated triterpenoid—**12** (200 or 500 nM) decreased the surface area covered by cell colonies 2.1- and 2.4-fold, respectively, compared to the control. An inhibitory effect of **12** on the clonogenic growth of U87 cells was also detected at 50 and 100 nM; however, this decrease was statistically insignificant (Figure 7A). In the case of U118 cells, 1.4- and 2-fold reductions in the area of cell colonies compared to the control were observed for concentrations of **12** at 200 and 500 nM, respectively (Figure 7A).

It is known that high cell motility is a key characteristic of metastatic cells [75] and that pentacyclic triterpenoids can effectively inhibit this process [56]. To evaluate the ability of tryptamide **12** to suppress the migration potential of glioblastoma cells, two independent approaches were undertaken, including assessment of cell movement on a 2D surface (scratch assay) and through a trans-well membrane (CIM-Plate). As shown in Figure 7B, the treatment of U118 cells with **12** at 100 nM for 48 h effectively inhibited 2D migration of the cells by two times compared to the control. The results for the trans-well assay independently confirmed the anti-motility activity of **12**—it was found that **12** used at 100 nM significantly suppressed the ability of U118 cells to cross the porous membrane of the CIM-Plate, starting already after 20 h of the treatment (Figure 7C).

Considering that vasculogenic mimicry (VM), a process whereby tumor cells form tumor cell-lined vasculatures to reinforce blood supply to cancerous tissue, plays a crucial role in glioblastoma progression and aggressiveness [76] and that some pentacyclic triterpenoids can inhibit this process [77,78], next we studied whether **12** affects VM in glioblastoma cells. It was found that treatment of U87 cells seeded on Matrigel-coated wells with **12** significantly decreased the numbers of tubular structures 2.2-fold compared with untreated cells (Figure 7D).

Thus, our results indicate a pronounced antitumor potential of **12** not only at toxic but also at non-toxic concentrations, leading to the suppression of clonogenic, migration and VM activities of glioblastoma cells.

#### 2.2.8. Compound **12** Displayed High Anti-Glioblastoma Potential In Vivo

##### Soloxolone Tryptamide **12** Inhibited the Growth of U87 Glioblastoma in a Murine Xenograft Model

Given the high antitumor potential of **12** with respect to glioblastoma cells observed in vitro, we further questioned whether the investigated triterpenoid suppresses the growth of glioblastoma in a murine model. To understand this, **12** was intraperitoneally injected as a solution in 10% Tween-80 at a dosage of 20 mg/kg into nude mice bearing subcutaneously implanted human U87 glioblastomas (the experimental setup is presented in Figure 8A). It was found that seven injections of **12** effectively suppressed glioblastoma growth, reaching a 3- and 4.4-fold decrease in the tumor volume on day 21 after tumor transplantation compared to the control and vehicle-treated group, respectively (Figure 8B). Surprisingly, the vehicle (10% Tween-80) stimulated tumor growth compared with the untreated control starting on day 16 after cell inoculation (Figure 8B,C), which can be explained by the known ability of Tween-80 to enhance the proliferative capacity of cells. Previously, Viennois et al. reported that injections of Tween-80 significantly increased the growth of tumor cells in colitis-associated colon cancer in vivo [79] and an addition of Tween-80 to a range of biocompatible materials was found to markedly stimulate cell proliferation [80,81]. Analysis of tumor weights also confirmed the high antitumor activity of **12**, which decreased this parameter by 2.1- and 3.4-fold compared to the untreated and vehicle-treated groups, respectively (Figure 8D). The treatment of glioblastoma-bearing mice with **12** did not induce severe adverse reactions during the experiment, since evaluation of organ indexes in the experimental and control groups did not reveal significant differences (Figure 8E).

Histologically, tumor nodes of U87 glioblastomas were represented by cells of various sizes and shapes (round, oval and bean-shaped), slender cytoplasm weakly stained with eosin and hyperchromic nuclei with moderate chromatin content (Figure 6F). Additionally, tumor tissues contained a large number of malformed vessels with thin walls (Figure 8F). The calculation of the numerical densities of mitotic (Nv) and Ki-67-positive cells, reflecting the aggressiveness and proliferative potential of the tumors, revealed a significant 4- and 2.4-fold decrease in mice treated with **12** compared with the vehicle-treated animals and a 3.4- and 3.5-fold decrease compared to the untreated control group, respectively (Figure 8H,I).

Finally, the HPLC–MS/MS analysis demonstrated that administration of **12** at a dosage of 20 mg/kg resulted in its accumulation in the brain tissue of tumor-bearing mice at a concentration of 118 ng/g (Figure 8J). The obtained results clearly confirm the ability of **12** to successfully penetrate the BBB and that the observed brain accumulation of **12** is dose-dependent (compare Figure 8J and Figure 3E).

Thus, our results demonstrated that administration of **12** in U87 glioblastoma-bearing mice led to significant retardation of primary tumor growth and a reduction in the proliferative potential of the tumor without systemic toxic effects.

##### Compound **12** Improved the Tumor Microenvironment in a Subcutaneous U87 Glioblastoma Xenograft Model

The tumor microenvironment is highly relevant to glioblastoma progression, playing an important role in the proliferation and invasion of tumor cells as well as their response to chemotherapy [82,83]. Some of the important factors of the tumor microenvironment influencing tumor growth are components of the extracellular matrix (ECM); for instance, overproduction of collagen and its accumulation within tumor stroma were found to enhance the malignancy of tumor cells [84] and limit the ability of chemotherapeutic drugs to diffuse and penetrate tumors [85]. In the case of glioblastoma, redundant tissue stiffness instigated by increased secretion and bundling of ECM proteins such as collagens and fibronectin is strongly associated with high proliferation, invasion and stemness of tumor cells—key factors in glioblastoma therapy resistance [86].

To evaluate whether **12** affects collagen expression in tumor tissue, Masson’s trichrome histochemical staining of tumor sections from control and experimental mice was performed. Representative histological images of U87 glioblastoma demonstrate that **12** decreased the collagen fiber content in tumor tissue compared with both untreated and vehicle-treated mice (Figure 9). Since pharmacological inhibition of collagen synthesis is a known approach to enhance drug penetration through tumor tissue and increase the sensitivity of tumor cells to chemotherapy [87], we suggest that the compound **12**-induced reduction observed in the stromal collagen distribution within U87 glioblastoma tissue is a favorable prognostic sign indicating the slowing down of tumor malignancy and progression and the formation of conditions conducive to more effective chemotherapy.

It is known that angiogenesis plays a crucial role in tumor growth, supplying tumor cells with oxygen and nutrients. Tumor vasculature is characterized by a tortuous morphology with poorly organized immature vessels and enhanced permeability, which leads to the formation of areas of persisting hypoxia and increased interstitial pressure, associated with aggressive tumor growth and impaired therapeutic delivery, respectively [88]. Despite the successful application of anti-angiogenic drugs in various treatment schemes for oncologic patients, wide-ranging blockade of neovascularization by anti-angiogenic molecules can lead to a hypoxic tumor microenvironment followed by the enhancement of tumor invasiveness, which limits the effectiveness of conventional chemotherapy [89]. In line with this fact, numerous clinical studies on glioblastoma have demonstrated that anti-angiogenic treatment can prolong progression-free survival only and that it does not not improve overall survival [88]. One of the novel promising approaches to target tumor vasculature is tumor vessel normalization—the formation of mature blood vessels with increased pericyte coverage, which alleviates hypoxia, significantly increases the delivery of chemotherapeutic drugs to tumor tissue and assists the function of antitumor immune cells [90]. Considering the ability of **12** to suppress vasculogenic mimicry (Figure 7D) and thus play an important role in the formation of poorly organized blood vessel networks in tumors [91], it was interesting to evaluate whether **12** influences vessel normalization in glioblastoma sites in vivo. Immunohistochemical staining of tumor sections of the control and experimental groups with anti-CD31 (endothelial cells) and anti-α-SMA (pericytes) primary antibodies showed that the administration of **12** increased the expression of both CD31 and α-SMA in the tumor tissue compared with the control and vehicle-treated mice, which indicates the enhancement of tumor vessel maturity caused by **12** (Figure 9). To the best of our knowledge, these data for the first time demonstrate that pentacyclic triterpenoid can stimulate tumor vessel normalization. Similar effects were previously found only for diterpenoid oridonin [92] and tetracyclic triterpenoid AECHL-1 [93].

Taken together, our findings clearly demonstrate that **12** displays significant anti-glioblastoma activity in vivo not only by direct inhibition of tumor cell proliferation but also by depletion of collagen content and normalization of blood vessels in tumor tissue— two processes mediating the high susceptibility of tumors to chemotherapy. Thus, the obtained results encourage further investigation of **12** as a component of combined chemotherapy in the treatment of glioblastoma together with conventional antitumor agents.

## 3. Materials and Methods

### 3.1. Chemistry

#### 3.1.1. General Experimental Procedures and Reagents

Elemental analyses were carried out on an Automatic CHNS-analyser EURO EA3000. Analyses indicated by the symbols of the elements were within ±0.4% of the theoretical values. Melting points were determined on a METTLER TOLEDO FP900 thermosystem and were uncorrected. The elemental composition of the products was determined from high-resolution mass spectra recorded on a DFS (double-focusing sector) Thermo Electron Corporation instrument. Hydrogen-1 and Carbon-13 NMR spectra were measured on Bruker spectrometers: DRX-500 (500.13 MHz for ^1^H and 125.76 MHz for ^13^C), AV-400 (400.13 MHz for ^1^H and 100.61 MHz for ^13^C) and AV-300 (300.13 MHz for ^1^H and 75.47 MHz for ^13^C). The solutions of each compound were prepared in CDCl_3_. Chemical shifts were recorded in *δ* (ppm), using *δ* 7.24 (^1^H NMR) and *δ* 76.90 (^13^C NMR) of CHCl_3_ as internal standards. Chemical shift measurements were given in ppm and the coupling constants (*J*) in hertz (Hz). The structures of the compounds were determined by NMR using standard one-dimensional and two-dimensional procedures (^1^H-^1^H COSY, ^1^H-^13^C HMBC/HSQC, ^13^C-^1^H HETCOR/COLOC). The purity of the final compounds and intermediates for biological testing was >95% as determined by HPLC analysis. HPLC analyses were carried out on a MilichromA-02, using a ProntoSIL 120-5-C18 AQ column (BISCHOFF, 2.0 × 75 mm column, grain size 5.0 lm). The mobile phase was Millipore-purified water with 0.1% trifluoroacetic acid at a flow rate of 150 µL/min at 35 °C with UV detection at 210, 220, 240, 260 and 280 nm. A typical run time was 25 min with a linear gradient of 0–100% methanol. Flash column chromatography was performed with silica gel (Merck, Kenilworth, NJ, USA, 60–200 mesh). All courses of all reactions were monitored by TLC analysis using Merck 60 F254 silica gel on aluminum sheets with the eluent CHCl_3_–MeOH (25:1.5 *v*/*v*).

Soloxolone methyl was prepared according to a previously reported method [28]. *N*,*N*′-Carbonyldiimidazole (CDI), *N*,*N*-Dimethylethylenediamine (99%), *N*,*N*-dimethyl-1,3-propanediamine and tryptamine (98%) and 3-aminopyridine (99%) were purchased from ACROS organics. Aminoethanol (99%) and 2-methoxyethylamine (99%) were purchased from Sigma Aldrich, 3-Aminopropanol (99%) was purchased from Alfa Aesar and 2-Aminoethanol was purchased from Ekos-1. All solvents used in the reactions were purified and dried according to previously reported procedures.

##### General Procedure A for Compounds (**2**–**5**, **8**–**12**)

Soloxolone (**1**) (1 equiv.) and CDI (1.2 equiv.) were dissolved in dry CH_2_Cl_2_ and stirred for 2.5 h at room temperature. Then, the corresponding amine (1.2 equiv.) was added to the solution and the reaction mixture was stirred at 35 °C. The reaction course was monitored by TLC (CHCl_3_–MeOH, 25–0.5). Upon reaching full conversion, the reaction mixture was diluted with CH_2_Cl_2_ and Et_2_O, washed sequentially with H_2_O and brine, and dried over anhydrous MgSO_4_. The solvent was evaporated to dryness.

##### General Procedure B for Compounds (**6**–**7**)

Soloxolone (**1**) (1 equiv.) and CDI (1.2 equiv.) were dissolved in dry CH_2_Cl_2_ and stirred for 2.5 h at room temperature. Then, the corresponding amine (1.2 equiv.) was added to the solution and the reaction mixture was stirred at 35 °C. The reaction course was monitored by TLC (CHCl_3_–MeOH, 25–0.5). Upon reaching full conversion, the solvent was evaporated to dryness.

#### 3.1.2. 2-Cyano-3,12-dioxo-18βH-olean-9(11),1(2)-dien-30-oic acid (Soloxolone) (**1**)

A solution of LiOH·H_2_O (4.0 g, 95 mmol) in H_2_O (30 mL) was added at 70 °C to a mixture of Soloxolone methyl (4.0 g, 7.9 mmol) in EtOH (80 mL). The mixture was stirred for 24 h, the reaction course was monitored by TLC. Then, the resulting mixture was concentrated under reduced pressure, diluted with a mixture of CH_2_Cl_2_/Et_2_O (1:2 *v*/*v*) (50 mL) and 5% aqueous hydrochloric acid solution was added until pH < 7. The organic layer was separated, the aqueous layer was extracted with CH_2_Cl_2_/Et_2_O. The combined organic layers were washed with brine and dried over MgSO_4_. The crude product (3.7 g) was purified by flash column chromatography (silica gel, 0–5% MeOH in CHCl_3_) to yield soloxolone (**1**) (2.3 g, 60%).

Hydrogen-1 and Carbon-13 NMR spectral data for soloxolone (**1**) were in good agreement with those in the literature [36].

#### 3.1.3. N-(2′-Hydroxyethyl)-2-cyano-3,12-dioxo-18βH-olean-9(11),1(2)-dien-30-amide (**2**)

Crude product **2** (400 mg) was obtained as a pale yellow amorphous solid according to the general procedure A from soloxolone (**1**) (400 mg, 0.81 mmol), dry CH_2_Cl_2_ (20 mL), CDI (158 mg, 0.98 mmol) and 2-aminoethanol (60 mg, 0.98 mmol). The crude product was purified by flash column chromatography (silica gel, 50–75% AcOEt in *n*-hexane) to yield compound **2** (293 mg, 55%) as a white amorphous solid. Elemental analysis: calculated 74.12% C, 8.67% H, 5.24% N, 11.97% O; found 73.82% C, 8.63% H, 5.20% N. Mp 152.7 °C (decomposition); ^1^H NMR (CDCl_3_, 500 MHz): *δ* =8.01 (s, 1H, H-1), 6.88 (br t, 1H, C(O)N*H*), 6.00 (s, 1H, H-11), 3.62–3.75 (m, 3H, [3.72 t, *J*_1′,2′_ = 4.4, 2H-1′], H-2′), 3.21–3.38 (m, 2H, [3.31 br s, O*H*], H-2′), 3.06 (d, 1H, *J* = 4.3, H-13), 0.80–2.10 (m, 37H, [1.47 (s, 3H, CH_3_-25)), 1.46 (s, 3H, CH_3_-26), 1.18 (s, 3H, CH_3_-23), 1.14 (s, 3H, CH_3_-24), 1.07 (s, 3H, CH_3_-29), 0.94 (s, 3H, CH_3_-27), 0.88 (s, 3H, CH_3_-28)]); ^13^C NMR (CDCl_3_, 125 MHz): *δ* =202.07 (s, C-12), 196.28 (s, C-3), 177.42 (s, C-30), 171.06 (s, C-9), 165.27 (d, C-1), 123.42 (d, C-11), 114.39 (c, C-2), 114.14 (s, *C*N), 62.47 (t, C-1′), 48.17 (d, C-13), 47.33 (d, C-5), 45.98 (s, C-8), 44.79 (s, C-4), 43.64 (s, C-20), 42.54 (t, C-2′), 42.48 (s, C-10), 42.23 (s, C-14), 37.51 (t, C-22), 37.47 (d, C-18), 33.68 (t, C-19), 31.94 (s, C-17), 31.36 (t, C-7, C-21), 29.22 (q, C-29), 26.76 (q, C-28), 26.69 (q, C-23), 26.45 (q, C-25), 26.05 (t, C-15), 25.80 (t, C-16), 24.47 (q, C-26), 21.50 (q, C-24), 21.33 (q, C-27), 17.93 (t, C-6).

#### 3.1.4. N-(3′-Hydroxypropyl)-2-cyano-3,12-dioxo-18βH-olean-9(11),1(2)-dien-30-amide (**3**)

Crude product **3** (310 mg) was obtained as a pale yellow amorphous solid according to the general procedure A from soloxolone (**1**) (300 mg, 0.73 mmol), dry CH_2_Cl_2_ (15 mL), CDI (159 mg, 0.98 mmol) and 3-aminopropanol (55 mg, 0.98 mmol). The crude product was purified by flash column chromatography (silica gel, 50–75% AcOEt in *n*-hexane) to yield compound **3** (137 mg, 55%) as a white amorphous solid. Elemental analysis: calculated 74.42% C, 8.82% H, 5.10% N, 11.66% O; found 74.68% C, 8.78% H, 5.15% N. Mp 150.8 °C (decomposition); ^1^H NMR (CDCl_3_, 500 MHz): *δ* = 8.00 (s, 1H, H-1), 7.00 (br t, 1H, C(O)N*H*), 5.99 (s, 1H, H-11), 3.54–3.66 (m, 3H, 2H-1′, H-3′), 3.41 (m, 1H, H-3′), 3.05–3.13 (m, 2H, [3.08 d, *J* = 4.4, H-13], O*H*), 0.80–2.10 (m, 39H, [1.49 (s, 6H, CH_3_-25, CH_3_-26), 1.24 (s, 3H, CH_3_-23), 1.15 (s, 3H, CH_3_-24), 1.05 (s, 3H, CH_3_-29), 0.96 (s, 3H, CH_3_-27), 0.92 (s, 3H, CH_3_-28)]); ^13^C NMR (CDCl_3_, 125 MHz): *δ* =201.53 (s, C-12), 196.21 (s, C-3), 178.12 (s, C-30), 170.70 (s, C-9), 165.11 (d, C-1), 123.38 (d, C-11), 114.52 (c, C-2), 114.14 (s, *C*N), 58.60 (t, C-1′), 48.07 (d, C-13), 47.44 (d, C-5), 46.00 (s, C-8), 44.84 (s, C-4), 43.89 (s, C-20), 42.53 (s, C-10), 42.30 (s, C-14), 37.69 (t, C-22), 37.45 (d, C-18), 35.65 (t, C-3′), 33.14 (t, C-19), 32.25 (t, C-2′), 31.97 (s, C-17), 31.38 (t, C-7, C-21), 29.51 (q, C-29), 26.80 (q, C-28), 26.75 (q, C-23), 26.54 (q, C-25), 26.09 (t, C-15), 25.85 (t, C-16), 24.55 (q, C-26), 21.45 (q, C-24), 21.36 (q, C-27), 17.99 (t, C-6).

#### 3.1.5. N-(5′-Hydroxypentyl)-2-cyano-3,12-dioxo-18βH-olean-9(11),1(2)-dien-30-amide (**4**)

Crude product **4** (310 mg) was obtained as a pale yellow amorphous solid according to the general procedure A from soloxolone (**1**) (300 mg, 0.73 mmol), dry CH_2_Cl_2_ (15 mL), CDI (159 mg, 0.98 mmol) and 5-aminopentanol (55 mg, 0.98 mmol). The crude product was purified by flash column chromatography (silica gel, 50–75% AcOEt in *n*-hexane) to yield compound **4** (137 mg, 55%) as a white amorphous solid. Elemental analysis: calculated 74.96% C, 9.09% H, 4.86% N, 11.10% O; found 75.01% C, 9.02% H, 4.88% N. Mp 115.7 °C (decomposition); ^1^H NMR (CDCl_3_, 500 MHz): *δ* =8.03 (s, 1H, H-1), 6.67 (br t, 1H, C(O)NH), 5.99 (s, 1H, H-11), 3.60 (t, 2H, *J_1′,2′_* = 6.0, 2H-1′), 3.23 and 3.26 (m, 2H, 2H-5′), 3.06 (d, 1H, *J* = 4.4, H-13) 2.98 (br s, 1H, OH), 0.80–2.15 (m, 43H, [1.47 (s, 3H, CH_3_-25)), 1.46 (s, 3H, CH_3_-26), 1.20 (s, 3H, CH_3_-23), 1.13 (s, 3H, CH_3_-24), 1.04 (s, 3H, CH_3_-29), 0.94 (s, 3H, CH_3_-27), 0.88 (s, 3H, CH_3_-28)]); ^13^C NMR (CDCl_3_, 125 MHz): *δ* =201.78 (s, C-12), 196.32 (s, C-3), 176.52 (s, C-30), 170.80 (s, C-9), 165.36 (d, C-1), 123.33 (d, C-11), 114.40 (c, C-2), 114.13 (s, CN), 62.10 (t, C-1′), 48.06 (d, C-13), 47.37 (d, C-5), 46.00 (s, C-8), 44.81 (s, C-4), 43.75 (s, C-20), 42.55 (s, C-10), 42.27 (s, C-14), 39.22, 37.76 (t, C-22), 37.40 (d, C-18), 33.03, 31.96, 31.93, 31.45, 31.36,29.59 (q, C-29), 28.66, 26.81 (q, C-28), 26.73 (q, C-23), 26.47 (q, C-25), 26.05 (t, C-15), 25.66 (t, C-16), 24.53 (q, C-26), 22.74 (t, C-3′) 21.51 (q, C-24), 21.35 (q, C-27), 17.96 (t, C-6).

#### 3.1.6. N-(2′-Methoxyethyl)-2-cyano-3,12-dioxo-18βH-olean-9(11),1(2)-dien-30-amide (**5**)

Crude product **5** (100 mg) was obtained as a pale yellow amorphous solid according to the general procedure A from soloxolone (**1**) (100 mg, 0.20 mmol), dry CH_2_Cl_2_ (5 mL), CDI (40 mg, 0.24 mmol) and 2-methoxyethanamine (18 mg, 0.24 mmol). The crude product was purified by flash column chromatography (silica gel, 0–10% AcOEt in CH_2_Cl_2_) to yield compound **5** (60 mg, 55%) as a white amorphous solid. Elemental analysis: calculated 74.42% C, 8.82% H, 5.10% N, 11.66% O; found 74.30% C, 8.51% H, 5.10% N. Mp 115.9 °C (decomposition); ^1^H NMR (CDCl_3_, 400 MHz): *δ* =8.00 (s, 1H, H-1), 6.79 (br t, 1H, C(O)N*H*), 5.99 (s, 1H, H-11), 3.45–3.6 (m, 4H, 2H-1′, 2H-2′), 3.35 (c, 3H, O*CH*_3_), 3.05 (d, 1H, *J* = 4.6, H-13), 0.80–2.20 (m, 37H, [1.47 (s, 3H, CH_3_-25)), 1.46 (s, 3H, CH_3_-26), 1.21 (s, 3H, CH_3_-23), 1.13 (s, 3H, CH_3_-24), 1.05 (s, 3H, CH_3_-29), 0.94 (s, 3H, CH_3_-27), 0.88 (s, 3H, CH_3_-28)]); ^13^C NMR (CDCl_3_, 100 MHz): *δ* =200.92 (s, C-12), 196.32 (s, C-3), 176.42 (s, C-30), 169.70 (s, C-9), 165.39 (d, C-1), 123.57 (d, C-11), 114.39 (s, C-2), 114.18 (s, *C*N), 70.85 (t, C-1′), 58.41 (q, O*CH*_3_), 47.98 (d, C-13), 47.44 (d, C-5), 45.92 (s, C-8), 44.83 (s, C-4), 43.78 (s, C-20), 42.47 (s, C-10), 42.25 (s, C-14), 39.02 (t, C-22), 37.86 (t, C-19), 37.27 (d, C-18), 33.06 (t, C-7*), 31.09 (t, C-21*), 31.61 (s, C-17), 31.39 (t, C-7, C-21), 29.55 (q, C-29), 26.81 (q, C-28), 26.74 (q, C-23), 26.50 (q, C-25), 26.10 (t, C-15), 25.70 (t, C-16), 24.57 (q, C-26), 21.55 (q, C-24), 21.36 (q, C-27), 18.01 (t, C-6).

#### 3.1.7. N-(2′-(Dimethylamino)ethyl)-2-Cyano-3,12-dioxo-18βH-olean-9(11),1(2)-dien-30-amide (**6**)

Crude product **6** was obtained as a pale yellow solid according to the general procedure B from soloxolone (**1**) (140 mg, 0.29 mmol), dry CH_2_Cl_2_ (10 mL), CDI (51 mg, 0.31 mmol) and *N*^1^,*N*^1^-dimethylethane-1,2-diamine (28 mg, 0.31 mmol). The crude product was purified by flash column chromatography (silica gel, 0–25% MeOH in CH_2_Cl_2_) to yield compound **6** (45 mg, 28%) as a colorless viscous liquid. HRMS: calculated for (C_35_H_51_O_3_N_3_)^+^ m/z = 561.3930; found m/z = 561.3932; ^1^H NMR (CDCl_3_, 400 MHz): *δ* = 8.02 (s, 1H, H-1), 6.80 (br t, 1H, C(O)N*H*), 6.00 (s, 1H, H-11), 3.44 (m, 2H, 2H-2′), 3.05 (d, 1H, *J* = 4.5, H-13), 2.54 (m, 2H, 2H-1′), 2.30 (s, 6H, N(*CH_3_*)_2_), 0.80–2.15 (m, 37H, [1.49 (s, 3H, CH_3_-25), 1.48 (s, 3H, CH_3_-26), 1.23 (s, 3H, CH_3_-23), 1.14 (s, 3H, CH_3_-24), 1.07 (s, 3H, CH_3_-29), 0.96 (s, 3H, CH_3_-27), 0.90 (s, 3H, CH_3_-28)]); ^13^C NMR (CDCl_3_, 125 MHz): *δ* =201.07 (s, C-12), 196.30 (s, C-3), 176.42 (s, C-30), 170.06 (s, C-9), 165.30 (d, C-1), 123.42 (d, C-11), 114.39 (c, C-2), 114.18 (s, *C*N), 57.67 (t, C-1′), 48.00 (d, C-13), 47.40 (d, C-5), 45.93 (s, C-8), 45.00 (q, N(*CH*_3_)_2_), 44.80 (s, C-4), 43.70 (s, C-20), 42.48 (s, C-10), 42.23 (s, C-14), 36.32 (t, C-2′), 37.51, 37.47 (d, C-18), 33.68, 31.94, 31.36, 29.50 (q, C-29), 26.80 (q, C-28), 26.70 (q, C-23), 26.50 (q, C-25), 26.07 (t, C-15), 25.80 (t, C-16), 24.50 (q, C-26), 21.55 (q, C-24), 21.35 (q, C-27), 17.98 (t, C-6).

#### 3.1.8. N-(3′-(Dimethylamino)propyl)-2-cyano-3,12-dioxo-18βH-olean-9(11),1(2)-dien-30-amide (**7**)

Crude product (**7)** was obtained as a pale yellow amorphous solid according to the general procedure B from soloxolone (**1**) (157 mg, 0.32 mmol), dry CH_2_Cl_2_ (10 mL), CDI (57 mg, 0.35 mmol) and *N*^1^,*N*^1^-dimethylpropane-1,3-diamine (36 mg, 0.35 mmol). The crude product was purified by flash column chromatography (silica gel, 0–25% MeOH in CH_2_Cl_2_) to yield compound **7** (62 mg, 34%) as a colorless viscous liquid. HRMS: calculated for (C_36_H_53_O_3_N_3_)^+^ m/z = 575.4087; found m/z = 547.4088; ^1^H NMR (CDCl_3_, 400 MHz): *δ* =8.00 (s, 1H, H-1), 6.80 (br t, 1H, C(O)N*H*), 6.00 (s, 1H, H-11), 3.30 (m, 2H, 2H-3′), 3.06 (d, *J* = 4.5, H-13), 2.56 (m, 2H, 2H-1′), 2.29 (s, 6H, N(*CH_3_*)_2_), 0.80–2.15 (m, 39H, [1.48 (s, 3H, CH_3_-25)), 1.47 (s, 3H, CH_3_-26), 1.22 (s, 3H, CH_3_-23), 1.14 (s, 3H, CH_3_-24), 1.05 (s, 3H, CH_3_-29), 0.95 (s, 3H, CH_3_-27), 0.90 (s, 3H, CH_3_-28)]); ^13^C NMR (CDCl_3_, 125 MHz): *δ* =201.05 (s, C-12), 196.35 (s, C-3), 176.44 (s, C-30), 170.02 (s, C-9), 165.41 (d, C-1), 123.44 (d, C-11), 114.39 (s, C-2), 114.21 (s, *C*N), 56.91 (t, C-1′), 47.99 (d, C-13), 47.40 (d, C-5), 45.93 (s, C-8), 44.93 (q, N(*CH*_3_)_2_), 44.83 (s, C-4), 43.73 (s, C-20), 42.48 (s, C-10), 42.23 (s, C-14), 37.81, 37.48 (t, C-3′), 37.44, 33.10, 31.95, 31.51, 31.37, 29.57 (q, C-29), 26.93 (t, C-2′), 26.88 (q, C-28), 26.73 (q, C-23), 26.51 (q, C-25), 26.08 (t, C-15), 25.73 (t, C-16), 24.58 (q, C-26), 21.55 (q, C-24), 21.37 (q, C-27), 18.01 (t, C-6).

#### 3.1.9. N-(4′-Bromophenyl)-2-Cyano-3,12-dioxo-18βH-olean-9(11),1(2)-dien-30-amide (**8**)

Crude product **8** (270 mg) was obtained as a pale yellow solid according to the general procedure A from soloxolone (**1**) (200 mg, 0.41 mmol), dry CH_2_Cl_2_ (10 mL), CDI (79 mg, 0.49 mmol) and 4-bromoaniline (84 mg, 0.49 mmol). The crude product was purified by flash column chromatography (silica gel, 0–10% AcOEt in CH_2_Cl_2_) to yield compound **8** (150 mg, 56%) as a white solid. Mp 283.0 °C (decomposition). HRMS: calculated for (C_37_H_45_^79^BrO_3_N_2_)^+^ m/z = 644.2614; found m/z = 644.2619; ^1^H NMR (CDCl_3_, 400 MHz): *δ* = 8.54 (br s, 1H, C(O)N*H*), 8.03 (s, 1H, H-1), 7.73 (d, 2H *J* = 8.9, H-2′, H-6′) and 7.40 (d, 2H *J* = 8.9, H-3′, H-5′)—*AB*, 6.08 (s, 1H, H-11), 3.13 (d, *J* = 4.6, H-13), 0.80–2.20 (m, 37H, [1.57 (s, 3H, CH_3_-25)), 1.40 (s, 3H, CH_3_-26), 1.25 (s, 3H, CH_3_-23), 1.18 (s, 3H, CH_3_-24), 1.16 (s, 3H, CH_3_-29), 1.01 (s, 3H, CH_3_-27), 0.91 (s, 3H, CH_3_-28)]); ^13^C NMR (CDCl_3_, 100 MHz): *δ* = 201.77 (s, C-12), 196.21 (s, C-3), 175.04 (s, C-30), 170.95 (s, C-9), 165.01 (d, C-1), 138.03 (s, C-1′), 131.63 (d, C-3′, C-5′), 123.34 (d, C-11), 121.11 (d, C-2′, C-6′), 115.95 (s, C-4′), 114.61 (s, C-2), 114.18 (s, *C*N), 48.12 (d, C-13), 47.53 (d, C-5), 46.14 (s, C-8), 45.16 (s, C-4), 44.91 (s, C-20), 42.66 (s, C-10), 42.45 (s, C-14), 37.71, 37.67, 37.71 (d, C-18), 33.38, 32.11, 31.67 (s, C-17), 31.46 29.39 (q, C-29), 26.75 (q, C-23, C-28), 26.48 (q, C-25), 26.25 (t, C-15), 25.79 (t, C-16), 24.59 (q, C-26), 21.63 (q, C-24), 21.43 (q, C-27), 18.07 (t, C-6).

#### 3.1.10. N-p-Tolyl-2-cyano-3,12-dioxo-18βH-olean-9(11),1(2)-dien-30-amide (**9**)

Crude product **8** (220 mg) was obtained as a pale yellow solid according to the general procedure A from soloxolone (**1**) (200 mg, 0.41 mmol), dry CH_2_Cl_2_ (10 mL), CDI (79 mg, 0.49 mmol) and p-toluidine (52 mg, 0.49 mmol). The crude product was purified by flash column chromatography (silica gel, 0–10% AcOEt in CH_2_Cl_2_) to yield compound **9** (94 mg, 40%) as a white solid. Mp 243.7 °C (decomposition). HRMS: calculated for (C_38_H_48_O_3_N_2_)^+^ m/z = 580.3665; found m/z = 580.3660; ^1^H NMR (CDCl_3_, 400 MHz): *δ* = 8.42 (br s, 1H, C(O)N*H*), 8.03 (s, 1H, H-1), 7.70 (d, 2H, *J* = 8.3, H-2′, H-6′) and 7.11 (d, 2H, *J* = 8.3, H-3′, H-5′)—*AB*, 6.08 (s, 1H, H-11), 3.13 (d, 1H, *J* = 4.6, H-13), 2.29 (s, 3H, CH_3_-7′)), 0.80–2.27 (m, 37H, [1.51 (s, 3H, CH_3_-25)), 1.50 (s, 3H, CH_3_-26), 1.25 (s, 3H, CH_3_-23), 1.18 (s, 3H, CH_3_-24), 1.16 (s, 3H, CH_3_-29), 1.01 (s, 3H, CH_3_-27), 0.90 (s, 3H, CH_3_-28)]); ^13^C NMR (CDCl_3_, 100 MHz): *δ* = 201.57 (s, C-12), 196.26 (s, C-3), 174.62 (s, C-30), 170.55 (s, C-9), 165.12 (d, C-1), 136.37 (s, C-4′), 132.99 (s, C-1′), 129.22 (d, C-3′, C-5′), 123.42 (d, C-11), 119.40 (d, C-2′, C-6′), 114.58 (s, C-2), 114.19 (s, *C*N), 48.11 (d, C-13), 47.51 (d, C-5), 46.10 (s, C-8), 44.99 (s, C-4), 44.91 (s, C-20), 42.61 (s, C-10), 42.41 (s, C-14), 37.78, 37.62 (d, C-18), 33.44, 31.73, 31.45, 29.54 (s, C-17), 29.39 (q, C-29), 26.75 (q, C-23, C-28), 26.49 (q, C-25), 26.24 (t, C-15), 25.72 (t, C-16), 24.60 (q, C-26), 21.69 (q, C-24), 21.43 (q, C-27), 20.72 (q, C-7′), (18.07 (t, C-6).

#### 3.1.11. N-(Pyridin-3-yl)-2-Cyano-3,12-dioxo-18βH-olean-9(11),1(2)-dien-30-amide (**10**)

Crude product **10** (220 mg) was obtained as a pale yellow solid according to the general procedure A from soloxolone (1) (200 mg, 0.41 mmol), dry CH_2_Cl_2_ (10 mL), CDI (79 mg, 0.49 mmol) and pyridin-3-amine (46 mg, 0.49 mmol). The crude product was purified by flash column chromatography (silica gel, 0–10% AcOEt in CHCl_3_) to yield compound **10** (120 mg, 52%) as a white solid. Mp 250 °C. HRMS: calculated for (C_36_H_45_O_3_N_3_)^+^ m/z = 567.3461; found m/z = 567.3461; ^1^H NMR (CDCl_3_, 300 MHz): *δ* =8.99 (d, 1H, *J* = 2.2, H-2′), 8.69 (br s, 1H, C(O)N*H*), 8.27 (m, 2H, H-4′,H-6′) 8.05 (s, 1H, H-1), 7.24 (m, 1H, H-5′), 6.08 (s, 1H, H-11), 3.11 (d, 1H, *J* = 4.6, H-13), 0.80–2.23 (m, 37H, [1.50 (s, 3H, CH_3_-25)), 1.49 (s, 3H, CH_3_-26), 1.22 (s, 3H, CH_3_-23), 1.18 (s, 3H, CH_3_-24), 1.13 (s, 3H, CH_3_-29), 0.98 (s, 3H, CH_3_-27), 0.89 (s, 3H, CH_3_-28)]); ^13^C NMR (CDCl_3_, 75 MHz): *δ* =201.68 (s, C-12), 196.24 (s, C-3), 175.63 (s, C-30), 170.95 (s, C-9), 165.15 (d, C-1), 143.96 (d, C-4′), 140.80 (d, C-6′), 135.81 (d, C-2′), 126.92 (d, C-5′), 123.41 (s, C-3′), 123.27 (d, C-11), 114.45 (s, C-2), 114.14 (s, *C*N), 48.00 (d, C-13), 47.42 (d, C-5), 46.05 (s, C-8), 45.10 (s, C-4), 44.84 (s, C-20), 42.62 (s, C-10), 42.41 (s, C-14), 37.70, 37.50, 33.21, 31.55, 31.36, 29.16 (s, C-17), 26.66 (q, C-29), 26.65 (q, C-23, C-28), 26.42 (q, C-25), 26.17 (t, C-15), 25.85 (t, C-16), 24.48 (q, C-26), 21.45 (q, C-24), 21.36 (q, C-27), 17.97 (t, C-6).

#### 3.1.12. N-(3′-(3′’,5′’-di-tert-butyl-4′’-hydroxyphenyl)propyl)-2-Cyano-3,12-dioxo-18βH-olean-9(11),1(2)-dien-30- amide (**11**)

Crude product **11** (440 mg) was obtained as a pale yellow amorphous solid according to the general procedure A from soloxolone (1) (300 mg, 0.61 mmol), dry CH_2_Cl_2_ (15 mL), CDI (118 mg, 0.73 mmol) and 4-(3-aminopropyl)-2,6-di-tert-butylphenol (193 mg, 0.73 mmol). The crude product was purified by flash column chromatography (silica gel, 0–10% AcOEt in CH_2_Cl_2_) to yield compound **11** (380 mg, 84%). Elemental analysis: calculated 78.22% C, 9.30% H, 3.80% N, 11.97% O; found 78.19% C, 9.30% H, 3.81% N. Mp 139.0 °C (decomposition); ^1^H NMR (CDCl_3_, 300 MHz): *δ* = 8.00 (s, 1H, H-1), 6.97 (s, 2H, H-2′ and H-6′), 6.77 (br t, 1H, C(O)N*H*), 6.00 (s, 1H, H-11), 5.04 (br s, 1H, O*H*), 3.47 (m, 1H) and 3.28 (m, 1H) (2H-9′), 3.07 (d, *J* = 4.3, H-13), 2.06 (m, 2H, 2H-7′), 0.80–2.20 (m, 57H, [1.50 (s, 3H, CH_3_-25)), 1.49 (s, 3H, CH_3_-26), 1.40 (s, 18H, C(*CH_3_)_3_*), 1.24 (s, 3H, CH_3_-23), 1.15 (s, 3H, CH_3_-24), 1.07 (s, 3H, CH_3_-29), 0.98 (s, 3H, CH_3_-27), 0.91 (s, 3H, CH_3_-28)]); ^13^C NMR (CDCl_3_, 75 MHz): *δ* = 201.07 (s, C-12), 196.24 (s, C-3), 176.31 (s, C-30), 170.15 (s, C-9), 165.18 (d, C-1), 151.58 (s, C-4′), 135.49 (s, C-3′ and C-5′), 132.27 (s, C-1′), 124.73 (d, C-2′ and C-6′), 123.34 (d, C-11), 114.41 (c, C-2), 114.14 (s, *C*N), 48.00 (d, C-13), 47.39 (d, C-5), 45.93 (s, C-8), 44.80 (s, C-4), 43.76 (s, C-20), 42.46 (t, C-2′), 42.46 (s, C-10), 42.27 (s, C-14), 39.40 (t, C-9′), 37.80 (t, C-22), 37.50 (d, C-18), 34.05, 33.31 (t, C-19), 33.02, 30.94 (s, C-17), 31.66 (t, C-7) 31.35 (t, C-21), 31.52, 30.13 (C(*CH_3_)_3_*), 29.53 (q, C-29), 26.87 (q, C-28), 26.67 (q, C-23), 26.42 (q, C-25), 26.08 (t, C-15), 25.71 (t, C-16), 24.47 (q, C-26), 21.47 (q, C-24), 21.32 (q, C-27), 17.97 (t, C-6).

#### 3.1.13. N-(2′-(1H-Indol-2-yl)-ethyl)-2-cyano-3,12-dioxo-18βH-olean-9(11),1(2)-dien-30-amide (**12**) 

Crude product **12** (340 mg) was obtained as a pale yellow amorphous solid according to the general procedure A from soloxolone (1) (170 mg, 0.35 mmol), dry CH_2_Cl_2_ (10 mL), CDI (67 mg, 0.42 mmol) and tryptamine (67 mg, 0.42 mmol). The reaction required careful monitoring by TLC due to its slow progress. The crude product was purified by flash column chromatography (silica gel, 40–60% AcOEt in *n*-hexane) to yield compound **12** (102 mg, 47%). Elemental analysis: calculated 77.69% C, 8.11% H, 6.63% N, 11.97% O; found 77.68% C, 8.09% H, 6.60% N. Mp 189.5–189.6 °C; ^1^H NMR (CDCl_3_, 400 MHz): *δ* =8.20 (br s, 1H, NH-1′), 8.03 (s, 1H, H-1), 7.60 (d, 1H, *J_5′,6′_* = 7.8, H-5′), 7.31 (d, 1H, *J_8′,7′_* = 8.1, H-8′), 6.95–7.2 (m, 4H, [7.13 s, H-2′], H-7′, H-6′, [6.98 br t, C(O)N*H*]), 5.97 (s, 1H, H-11), 3.83 (m, 1H) and 3.62 (m, 1H) (2H-11′), 3.1 (m, 2H, 2H-10′), 2.83 (d, 1H, *J* = 4.3, H-13), 0.80–2.20 (m, 37H, [1.50 (s, 3H, CH_3_-25)), 1.44 (s, 3H, CH_3_-26), 1.28 (s, 3H, CH_3_-29), 1.24 (s, 3H, CH_3_-23), 1.16 (s, 3H, CH_3_-24), 0.92 (s, 3H, CH_3_-27), 0.73 (s, 3H, CH_3_-28)]); ^13^C NMR (CDCl_3_, 100 MHz): *δ* = 201.1 (s, C-12), 196.33 (s, C-3), 176.88 (s, C-30), 169.91 (s, C-9), 165.33 (d, C-1), 136.14 (s, C-9′), 127.57 (s, C-4′), 123.34 (d, C-11), 121.81 (d, C-2′), 121.63 (d, C-7′), 118.86 (d, C-5′, C-6′), 114.52 (c, C-2), 114.26 (s, *C*N), 113.17 (s, C-3′), 110.91 (d, C-8′), 47.85 (d, C-13), 47.56 (d, C-5), 45.96 (s, C-8), 44.91 (s, C-4), 43.83 (s, C-20), 42.56 (s, C-10), 42.37 (s, C-14), 39.90 (t, C-10′), 37.69 (t, C-22), 37.04 (d, C-18), 32.98 (t, C-19), 31.94 (s, C-17), 31.48 (t, C-7*), 31.44 (t, C-21*), 29.26 (q, C-29), 26.77 (q, C-23), 26.53 (q, C-25), 26.43 (q, C-28), 26.18 (t, C-15), 23.11 (t, C-16), 25.00 (t, C-11′), 24.42 (q, C-26), 21.42 (q, C-24), 21.43 (q, C-27), 18.1 (t, C-6).

### 3.2. Biological Evaluations

#### 3.2.1. Cell lines and Novel Compounds

Human cervical carcinoma HeLa cells, human duodenal adenocarcinoma HuTu-80 cells, human glioblastoma U87 and U118 cells and mouse neuroblastoma Neuro2a cells obtained from the Russian Culture Collection (Institute of Cytology of the Russian Academy of Sciences (RAS), St. Petersburg, Russia) were cultured in Dulbecco’s modified Eagle’s medium (DMEM) (Sigma Aldrich, St. Louis, MI, USA). Human neuroblastoma KELLY cells purchased from the American Type Culture Collection (ATCC, Manassas, VA, USA) were cultured in Roswell Park Memorial Institute (RPMI-1640) medium (Sigma Aldrich, St. Louis, MI, USA) containing 1% (*v*/*v*) GlutaMAX (Gibco, Madison, WI, USA). Human non-transformed hFF3 foreskin fibroblasts were kindly provided by Dr. Olga Koval (Institute of Chemical Biology and Fundamental Medicine of the Siberian Branch of RAS (SB RAS), Novosibirsk, Russia) and were cultured in Iscove’s Modified Dulbecco’s Medium (IMDM) (Sigma Aldrich, USA). All culture media contained 10% (*v*/*v*) heat-inactivated fetal bovine serum (FBS) (Gibco, USA) and antibiotic–antimycotic solution containing penicillin at 10,000 IU/mL, streptomycin at 10,000 µg/mL and amphotericin at 25 µg/mL (MP Biomedicals, Illkirch-Graffenstaden, France). The cells were incubated at 37 °C in humidified 5% CO_2_-containing air atmosphere (hereafter standard conditions). The novel compounds were dissolved in DMSO (stock solution: 10 mM) and stored at −20 °C before experiment.

#### 3.2.2. Mice

Six-to-eight-week-old female Balb/C nude mice with average weights of 22–25 g were purchased from the Center for Genetic Resources of Laboratory Animals at the Institute of Cytology and Genetics SB RAS (Novosibirsk, Russia). Animals were kept in plastic cages (6 animals per cage) under a normal daylight schedule in temperature-controlled, specific pathogen-free conditions. Water and food were provided ad libitum. All animal procedures were carried out in strict accordance with the European Communities Council Directive 86/609/CEE. The experimental protocol was approved by the Committee on the Ethics of Animal Experiments at the Institute of Cytology and Genetics SB RAS (protocol no. 56 from 10 August 2019).

#### 3.2.3. Evaluation of Cytotoxicity of Novel Compounds by MTT Assay

The cells in the exponential growth phase were seeded in quadruplicate in 96-well plates at a density of 1 × 10^4^ cells/well for HeLa, U87, U118, KELLY and Neuro2a cells; 1.5 × 10^4^ cells/well for HuTu-80 cells; 2 × 10^4^ cells/well for B16 cells; and 7 × 10^3^ cells/well for hFF3 fibroblasts. Cells were incubated under standard conditions overnight followed by their treatment with the investigated compounds (1–20 µM) for 24 h. Then, 10 µL of MTT solution was added to the cells to a final concentration of 0.5 mg/mL and plates were placed under standard conditions for 3 h followed by an aspiration of MTT-containing medium and a dissolution of formazan crystals with 100 µL of DMSO. The absorbance of each well was read at test and reference wavelengths of 570 and 620 nm, respectively, on a Multiscan RC plate reader (Thermo LabSystems, Helsinki, Finland). The IC_50_ values (the concentration of compound resulted in a decrease in A_570_ to 50% of A_570_^control^) were calculated by extrapolation of the dose–response curves.

#### 3.2.4. In Silico Prediction of the Blood–Brain Barrier (BBB) Permeability of the Novel Compounds and Their P-glycoprotein (P-gp) Substrate Specificity

The ability of the novel compounds to pass the BBB was predicted using the AlzPlatform (https://www.cbligand.org/AD/; accessed date: 20 March 2022) and PreADMET (https://preadmet.qsarhub.com/adme/; accessed date: 20 March 2022) chemoinformatics web tools. Doxorubicin (DOX) and temozolomide (TMZ) were used as reference molecules incapable and capable of penetrating the BBB, respectively. The threshold of BBB scores calculated by AlzPlatform and PreADMET, which indicate BBB-permeable agents, were ≥0.02 and ≥0.1, respectively. The ability of the novel compounds to act as substrates and inhibitors of P-glycoprotein was predicted by a panel of web tools, including ADMETlab (https://admet.scbdd.com/; accessed date: 20 March 2022), SwissADME (http://www.swissadme.ch/; accessed date: 20 March 2022), Vienna LiverTox Workspace (https://livertox.univie.ac.at/; accessed date: 20 March 2022) and PreADMET (https://preadmet.qsarhub.com/adme/; accessed date: 20 March 2022).

#### 3.2.5. Evaluation of Brain Accumulation of **12**

Six healthy Balb/C nude mice were i.p. administered with compound **12** in 10% Tween-80 at doses of 20 mg/kg or 50 mg/kg. The treatment was carried out three times a week. The total number of injections was seven. Mice were sacrificed 48 h after the last compound injection, and brain samples were collected for subsequent HPLC–MS/MS analysis.

#### 3.2.6. Working Solutions for Analysis of Brain Accumulation of **12**

The stock solution of **12** was prepared by dissolving 1.5–1.6 mg (exact amount) of the substance in a corresponding amount of 100% acetonitrile to obtain a concentration of 1.0 mg/mL. The working solutions of the compound with concentrations of 0.01, 0.02, 0.04, 0.08, 0.1, 0.2, 0.4, 0.8, 1, 1.6, 2, 4 and 10 µg/mL were prepared by dilution of the stock solution with acetonitrile. To obtain the working solution of the internal standard, an exact amount of 2,5-BDPO (Appendix A) was dissolved in methanol followed by a series of dilutions to obtain a concentration of 200 ng/mL. All solutions were stored at −18 °C. Before sample preparation, the solutions were brought to room temperature.

#### 3.2.7. Brain Tissue Homogenization Protocol

Mouse brain tissue samples were inserted in polypropylene tubes containing homogenization beads (Al_2_O_3_ and SiC particles, 1.6 mm, MP Biomedicals, Santa Ana, CA, USA). Four-hundred microliters of water per 100 mg of tissue sample was added followed by homogenization for 60 s at medium speed (Minilys, Bertin Technologies, Montigny-le-Bretonneux, France).

#### 3.2.8. Preparation of Calibrators and Experimental Samples for Quantification of **12** in Mouse Brain Tissues

To prepare calibrators, mouse brain tissue samples taken from three healthy untreated animals were pooled and homogenized as described above. To a 95 µL of brain homogenate, 5 µL of a working solution of **12** in acetonitrile was added and vortex-mixed. A series of samples containing 2.5, 5, 10, 25, 50, 100, 250, 500, 1000 and 2500 ng/g was prepared. The samples were held for 20–30 min at ambient temperature before their further processing. To a 100 µL of a homogenate sample, 400 µL of methanol was added and the sample was vortexed for 20–30 s. A solution of the internal standard (20 µL) was added followed by vortexing of the sample for an additional 20–30 s and shaking for 60 min at 1500 rpm in an orbital shaker. The samples were centrifuged for 10 min at 13,400 rpm (Eppendorf MiniSpin, Eppendorf, Germany). The obtained supernatants were transferred into vial inserts and analyzed. To quantify **12** in experimental animal brain tissue, a sample weighing c.a. 50–100 mg was taken and processed according to the protocol.

#### 3.2.9. HPLC–MS/MS Conditions

HPLC–MS/MS analysis was performed using a Shimadzu LC-20AD Prominence chromatograph equipped with a gradient pump, cooled autosampler and column oven. The chromatographic separation was performed on a column (2 × 75 mm, 5 µm) packed with reverse-phase sorbent ProntoSIL 120-5-C18AQ (Econova, Novosibirsk, Russia). Water containing 0.1% HCOOH was used as mobile phase A and methanol containing 0.1% HCOOH was used as mobile phase B. The gradient was as follows: 0 min—5% B; 0.5 min—5% B; 5.5 min—95% B; 7.5 min—95% B. The flow rate was 250 µL/min and the sample injection volume was 10 µL. Each sample was analyzed twice; after a run, the column was equilibrated for the next analysis.

Mass spectrometric detection was performed on a 6500 QTRAP mass spectrometer (AB SCIEX, USA) with the use of positive ESI. The following conditions were used for the analysis: detection mode—MRM, CUR (curtain gas) = 25 psi, CAD (collision gas activated dissociation) = Medium, IS (ion source voltage) = 5500 V, TEM (temperature) = 400 °C, GS1 (gas-sprayer) = 25 psi, GS2 (gas-dehydrator) = 25 psi, EP (entrance potential) = 10.0 V, dwell time = 150 ms. The parameters for the detection of compound **12** and 2,5-BDPO (IS) in the MRM mode are given in Appendix A. The device was controlled and information was collected using Analyst 1.6.3 (AB SCIEX, Framingham, MA, USA) software. Chromatograms were processed using MultiQuant 2.1 (AB SCIEX, USA) software.

#### 3.2.10. Analysis of Pro-Apoptotic Activity of **12**

To quantify the percentage of cells undergoing apoptosis, glioblastoma U87 and U118 cells were seeded in 6-well plates at 2.8 × 10^5^ cells/well in complete growth medium and incubated at standard conditions overnight followed by treatment with **12** at 4 µM for 24 h. Thereafter, the cells were harvested by trypsinization (TrypLE Express (Gibco, USA)) and centrifugation at 400 g for 5 min, washed with PBS and stained with 5 µL of Annexin V-APC (Sony Biotechnology Inc., San Jose, CA, USA) and 10 µL of propidium iodide (PI) (Invitrogen, Waltham, MA, USA) for 15 min at room temperature in the dark. After the incubation time, the cells were analyzed using a NovoCyte Flow Cytometer (ACEA Biosciences Inc., San Diego, CA, USA). For each sample, 10,000 events were acquired.

#### 3.2.11. Analysis of Caspase-3/-7 Activation

U87 cells were plated at a density of 2.8 × 10^5^ cells/well in 6-well plates and treated with **12** at 2 and 4 μM for 6 and 24 h. Then, the cells were collected by trypsinization and centrifugation, washed with PBS and stained with CellEvent™ Caspase-3/7 Green ReadyProbes™ Reagent (Molecular Probes, Eugene, OR, USA) according to the manufacturer’s manual. Briefly, harvested cells were resuspended in 200 μL of PBS containing the reagent (1 drop/mL) and incubated for 30 min at room temperature. After that, flow cytometry analysis of the cells was performed using the NovoCyte Flow Cytometer. In total, 10,000 events were acquired for each sample.

#### 3.2.12. Analysis of Autophagy-Inducing Activity of **12**

To evaluate the ability of **12** to induce autophagy, U87 and U118 cells were seeded at 2.8 × 10^5^ cells/well in 6-well plates and incubated under standard conditions overnight followed by their treatment with **12** in amounts of 2 and 4 μM for 24 h. Then, the cells were collected and washed with PBS as described above and resuspended in 500 µL of PBS containing monodancylcadaverine (MDC) at 50 μM. After incubation for 15 min under standard conditions, the cells were washed with PBS and analyzed by flow cytometry. For each sample, 10,000 events were acquired.

In order to evaluate whether autophagy plays an important role in compound **12**-induced glioblastoma cell death, U87 cells were seeded in quadruplicate at a density of 1 × 10^4^ cells/well in 96-well plates, incubated overnight under standard conditions and co-treated with **12** at 1–5 μM and chloroquine (CQ) at 50 μM for 24 h. Then, cell viability was measured using an MTT test and the curves of cytotoxicity of **12** in the presence or absence of CQ were compared.

#### 3.2.13. Analysis of Mitochondrial Membrane Potential

U87 cells were cultured in 6-well plates for 24 h, incubated overnight under standard conditions and treated with **12** at 4 μM for 6 and 24 h. Thereafter, the cells were harvested, resuspended at 10^6^ cell/mL in JC-1-containing PBS (5 µg/mL) and incubated under standard conditions for 30 min followed by the washing of the cells with PBS and flow cytometry analysis. In total, 10,000 events were acquired for each sample.

U118 cells plated in 6-well plates were treated with **12** at 4 μM for 6, 18 and 24 h and stained using a Mitochondrial Membrane Potential/Annexin V Apoptsis Kit (Molecular Probes, USA) according to the manufacturer’s instructions. Briefly, harvested cells were incubated in Annexin-binding buffer containing MitoTracker Red at 40 nM for 30 min under standard conditions. After that, the cells were washed with PBS, resuspended in Annexin-binding buffer containing 5 µL of Annexin V-Alexa Fluor 488 and incubated at room temperature for 15 min. After the incubation time, the cells were analyzed by flow cytometry. For each sample, 10,000 events were acquired.

#### 3.2.14. Evaluation of Mitochondrial Mass

U87 cells were seeded at a density of 2.8 × 10^5^ cells/well in 6-well plates and incubated overnight, followed by their treatment with **12** at 2 and 4 µM for 24 h. Next, the cells were collected and incubated in 200 µL of PBS containing MitoTracker Green at 100 nM for 30 min under standard conditions. Thereafter, the cells were washed twice with PBS and analyzed by flow cytometry. In total, 10,000 events were recorded for each sample.

#### 3.2.15. Analysis of ROS Production

Intracellular ROS levels in glioblastoma cells were measured using a DCFDA/H2DCFDA-Cellular ROS Assay Kit (Abcam, Cambridge, UK) according to the manufacturer’s instructions. Briefly, U87 cells were collected by trypsinization and stained with DCFDA at 20 μM for 30 min under standard conditions and seeded in 6-well plates at 2.8 × 10^5^ cells/well. Next, the cells were treated with **12** at 4 µM for 24 h, followed by their harvesting and flow cytometry analysis. In total, 10,000 events were recorded for each sample.

In order to understand whether ROS play a crucial role in compound **12**-induced cell death, U87 cells were plated in quadruplicate in 96-well plates at a density of 1 × 10^4^ cells/well and incubated in the presence of **12** at 4 µM and N-Acetyl-L-cysteine (NAC) at 2 µM for 24 h under standard conditions. Then, cell viability was measured using an MTT test.

#### 3.2.16. Molecular Docking

Molecular docking of **12** with LonP1 was carried out using Autodock Vina [94]. The 3D structure of LonP1 (Protein Data Bank (PDB) ID: 6X27) was uploaded from the Research Collaboratory for Structural Bioinformatics (RCSB) Protein Data Bank (https://www.rcsb.org/; accessed date: 1 April 2022). Co-crystalized ligand and water molecules were extracted from the PDB file of LonP1 and polar hydrogens as well as Gasteiger charges were added to the protein structure using AutoDock Tools 1.5.6. The 3D structure of **12** was created by Marvin Sketch 5.12 and its geometry was optimized with the MMFF94 force field using Avogadro 1.2.0. All torsions were allowed to rotate freely during docking. The grid box parameters were: center_x = 68.914, center_y = (−42.908), center_z = (−110.778); size_x = 40, size_y = 40, size_z = 40. The results of docking were imported and analyzed by BIOVIA Discovery Studio Visualizer 17.2.0.

#### 3.2.17. Colony Formation Assay

U118 cells were seeded in 96-well plates in six replicates at a density of 200 cells/well and treated with **12** at 25–500 nM for 14 days under standard conditions. The culture media were replaced with fresh media containing corresponding concentrations of **12** every 6 days. After the incubation time, cell colonies were fixed with 4% paraformaldehyde, stained with crystal violet dye (0.1% *w*/*v*) and photographed using the iBright 1500 Imaging System (Invitrogen, USA). The percentage of area covered by cell colonies was computed using the ColonyArea ImageJ plugin [95].

#### 3.2.18. Scratch Assay

To assess the 2D motility of glioblastoma cells, U118 cells were seeded in triplicate in 6-well plates at a density of 8 × 10^5^ cells/well and cultured to 80% confluence. Thereafter, the cell monolayers were scratched with 10 µL pipette tips and washed twice with PBS to remove floating cells and cell debris and treated with **12** at 100 nM for 48 h. At 0 h and 48 h, the scratched cell monolayers were imaged using a Primo Vert microscope with phase contrast optics (ZEISS, Oberkochen, Germany) equipped with a AxioCam ERc5s camera (ZEISS, Oberkochen, Germany). The wound closure rate was calculated using ImageJ software. In total, 4–5 regions on a single scratch were analyzed.

#### 3.2.19. Evaluation of Trans-Well Motility of Tumor Cells Using the xCELLigence Platform

To assess the trans-well motility of glioblastoma cells, U118 cells were seeded in quadruplicate at 2 × 10^4^ cells/well in the upper chamber of a CIM-Plate in FBS-free DMEM in the presence or absence of **12** at 100 nM. To induce trans-well motility of the cells, DMEM containing 10% FBS was placed in the lower chamber of the CIM-Plate. After that, the electrical impedance of the sensor electrodes mounted at the lower side of the porous membrane separating the upper and lower chambers of the CIM-plate was further measured using an xCELLigence RTCA DP instrument (ACEA Biosciences Inc., San Diego, CA, USA) every 1 h for 48 h.

#### 3.2.20. Vasculogenic Mimicry

A 96-well plate was placed on ice for pre-cooling and Matrigel^®®^ Matrix (Corning, Corning, NY, USA) was thawed at 4 °C and added to each well of the plate (50 μL/well). Then, the plate was placed at 37 °C in 5% CO_2_ for 1 h to solidify the Matrigel. U87 cells were seeded at 2 × 10^4^ cells/well on the Matrigel-coated wells and treated with **12** at 200 nM for 5 h. At the end of the incubation time, each well was analyzed under a microscope with a phase contrast Primo Vert (ZEISS, Germany). Tubular structures in each field were imaged with a AxioCam ERc5s camera (ZEISS, Germany) and the average number of tubules from three to four random fields in each well was quantified by ImageJ. The experiment was performed in triplicate.

#### 3.2.21. Tumor Transplantation and Design of Animal Experiments

A xenograft model of tumor progression was induced by subcutaneous (s.c.) injection of U87 glioblastoma cells (2 × 10^7^ cells/mL) suspended in 0.1 mL of saline buffer with 0.05 mL of Matrigel^®^ Matrix (Corning, NY, USA) into the left flank of Balb/C nude mice. On day 5 after tumor transplantation, the mice were divided into three groups (n = 6 per group): (1) mice without treatment (control); (2) mice that received intraperitoneal (i.p.) injections of 10% Tween-80 (vehicle); and (3) mice that received i.p. injections of **12** in 10% Tween-80 at a dose of 20 mg/kg. The treatment was carried out thrice per week. The total number of injections was seven. During the experiment, the tumor volumes were determined three times a week using caliper measurements and were calculated as *V*= (*D* × *d*^2^)/*2*, where *D* is the longest diameter of the tumor node and *d* is the shortest diameter of the tumor node perpendicular to *D*. The mice were sacrificed on day 21 after tumor transplantation and the tumors and brains were collected for subsequent histological and pharmacological analysis.

#### 3.2.22. Toxicity Assessment

During the experiment, the general status of the animals and body weight was monitored. At the end of the experiment, organs were collected and organ indexes were calculated as (organ weight/body weight) × 100%.

#### 3.2.23. Histology and Immunohistochemistry

For the histological study, the tumor specimens were fixed in 10% neutral-buffered formalin (BioVitrum, St. Petersburg, Russia), dehydrated in ascending ethanols and xylols and embedded in HISTOMIX paraffin (BioVitrum, St. Petersburg, Russia). The paraffin sections (5 μm) were sliced on a Microm HM 355S microtome (Thermo Fisher Scientific, Waltham, MA, USA) and stained with hematoxylin and eosin. Extracellular matrix deposition was determined using Masson’s trichrome staining.

For the immunohistochemical study, the tumor sections (3–4 μm) were deparaffinized and rehydrated. Antigen retrieval was carried out after exposure in a microwave oven at 700 W. The samples were incubated with anti-Ki-67 (ab16667, Abcam, Boston, MA, USA), anti-α-SMA (ab5694, Abcam, USA) or anti-CD31 (ab182981, Abcam, USA) primary antibodies according to the manufacturer’s protocol. Then, the sections were incubated with secondary horseradish peroxidase (HPR)-conjugated antibodies, exposed to the 3,3′-diaminobenzidine (DAB) substrate (Rabbit Specific HRP/DAB (ABC) Detection IHC Kit, ab 64261, Abcam, USA) and stained with Mayer’s hematoxylin.

All the images were examined and scanned using an Axiostar Plus microscope equipped with an Axiocam MRc5 digital camera (ZEISS, Germany) at magnifications of ×400.

Morphometric analysis of tumor sections included evaluation of the numerical density (Nv) of mitoses and Ki-67-positive cells in a square unit of tumor tissue—3.2 × 10^6^ μm^2^ in this case. At least ten random fields from the tumor specimens of six mice in each group (60 testing fields in total) were studied.

#### 3.2.24. Statistical Analysis

Statistical analysis was performed using the Microsoft Excel program, assuming a significance level of changes at *p* < 0.05. The unpaired Student’s *t*-test assessed statistical significance between the experimental groups and the control. 

## 4. Conclusions

In this study, eleven amides of soloxolone, the cyano enone derivative of 18βH-glycyrrhetinic acid, were designed and synthesized as antitumor candidates using **SM** as the starting compound. Screening of their cytotoxicity in a panel of tumor cell lines demonstrated the similar bioactivity of novel derivatives with SM (average IC_50_^soloxolone amides^ = 2.6 µM versus IC_50_**^SM^** = 1.6 µM). Further in silico analysis revealed that modifications can impart enhanced BBB permeability to novel compounds compared to unmodified **SM**, as was further verified in a murine model by HPLC–MS/MS analysis: soloxolone tryptamide **12** at a dosage of 50 mg/kg was found to reach a concentration of 490 ng/g in the brain tissue of nude mice receiving a widely used treatment scheme (dose: 50 mg/kg; i.p.; three times a week; seven injections in total). A cytotoxic assay demonstrated that the novel compounds effectively inhibited the viability of glioblastoma and neuroblastoma cells at low micromolar concentrations (average IC_50_ = 2.5 µM) and compound **12** was chosen as a hit compound to evaluate the mechanism of action. Mechanistic studies showed that **12** induced ROS-dependent and autophagy-independent death of glioblastoma cells associated with mitochondrial stress, probably driven by direct interaction of **12** with the active site of mitochondrial LonP1 protease, which led to the dissipation of Δψ_M_, abundant ROS generation and mitogenesis, with subsequent activation of caspase-mediated apoptosis. Evaluation of the bioactivities of **12** in non-toxic concentrations confirmed its marked antitumor potential: it was shown that **12** effectively inhibited the clonogenic activity and motility of glioblastoma cells as well as their vascular mimicry potency. Further animal experiments verified the high antitumor activity of **12**: it was found that **12** significantly inhibited the growth of U87 glioblastoma in a xenograft model, the effect mediated not only by direct blockage of tumor cell proliferation but also by depletion of collagen content and blood vessel normalization in tumor tissue. Thus, our findings clearly show that compound **12** can be considered a promising novel anti-glioblastoma drug candidate.

## Data Availability

Data is contained within the article and Appendix A.

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
