# Peer review of "Novel Soloxolone Amides as Potent Anti-Glioblastoma Candidates: Design, Synthesis, In Silico Analysis and Biological Activities In Vitro and In Vivo"

_pharmaceuticals, 2022, doi:10.3390/ph15050603_

Round 1

Reviewer 1 Report

This is a very nice manuscript holding all: a good conceptualization, experiment and biological evaluation. although the progress concerning cytotoxicity is not very high as compared to the parent compound, the introduction of an amide has - once again - proven useful. No changes/... required. Excellent!

Author Response

Dear Reviewer #1,

We sincerely thank you for your review of our manuscript and recommendation of this article for publication in Pharmacauticals.

Reviewer 2 Report

The authors have synthesized eleven amide derivatives of soloxolone with similar bioactivity as soloxolone methyl (SM). The introduction of the amide moiety may enhance the ability to pass the blood-brain barrier. According citotoxic assay the novel compounds inhibited the viability of the glioblastoma and neuroblastoma cells at low micromolar concentrations.

The compound 12 was chosen as a hit compound and its marked antitumor potential was confirmed in animal experiments too.

The manuscript is suitable for publication after minor revision.

The abbreviations should be explained, where they are mentioned at first, eg: line 59: CDDO-Et, line 60: CDDO-TEA, line 63: BBB, line 263: MRM, line 369: CDDO-Im, etc…

Scheme 1. The structures of Soloxolone methyl, Soloxolone (1) and compounds (2-12) are not correct, the carbon atom of the nitril group is connecting to the carbon atom C(2) of the „A” ring .

Chapter 2.1 Chemistry: The structure elucidation of the new compounds by NMR results is not detailed enough. HRMS results would be more informative instead of elemental analysis.

Lines 254, 520, 646, etc. nude mice, capital letters are not necessary.

Author Response

Dear Reviewer #2,

We are genuinely thankful to you for your careful analysis of our manuscript and highly valuable remarks. We revised the manuscript according to your comments and, please, let us respond to your questions.

  1. The abbreviations should be explained, where they are mentioned at first, eg: line 59: CDDO-Et, line 60: CDDO-TEA, line 63: BBB, line 263: MRM, line 369: CDDO-Im, etc…

Authors: Corrected. All the abbreviations were deciphered throughout the manuscript. The corrections introduced into the text are marked in yellow. Please, see: CDDO ethylamide (CDDO-Et) (line 59), CDDO 2,2,2-trifluoroethylamide (CDDO-TFEA) (lines 59-60), N,N-dimethylformamide (DMF) (line 116), nuclear magnetic resonance (NMR) (lines 134-135), 3-(4,5-dimethylthiazol-2-yl)-2,5-diphenyltetrazolium bromide (MTT) (line 148), quantitative structure activity relationship (QSAR) (lines 210-211), high performance liquid chromatography–tandem mass spectrometry (HPLC-MS/MS) (lines 265-266), multiple reaction monitoring (MRM) (lines 273-274), reactive oxygen species (ROS) (line 383), dichlorodihydrofluorescein diacetate (DCFDA) (line 448), Protein Data Bank (PDB) (line 470).

  1. Scheme 1. The structures of Soloxolone methyl, Soloxolone (1) and compounds (2-12) are not correct, the carbon atom of the nitril group is connecting to the carbon atom C(2) of the „A” ring .

Authors: Corrected. We are deeply grateful to the reviewer for carefully reading our manuscript and pointing out this significant inaccuracy in the structures. The Scheme 1 was corrected.

  1. Chapter 2.1 Chemistry: The structure elucidation of the new compounds by NMR results is not detailed enough. HRMS results would be more informative instead of elemental analysis.

Authors: According to the Instructions for the Authors of the journal Pharmaceuticals, «Reports on previously undescribed organic compounds should include, as supplementary data, 1H, 13C and/or other key heteronuclear or 2D NMR spectra, together with high-resolution mass spectrometry (HRMS) OR elemental analysis» 

The reactions to produce amides were carried out using commercially available amines and run smoothly without obtaining "unexpected" products, so we decided to opt for "elemental analysis" over HRMS. This method allows us not only to determine the elemental composition but also once again confirm the purity of the compounds (for example, the absence of significant amounts of the solvent that is used in chromatography). However, if the reviewer insists on HPLC, we are ready to provide these analyzes, however it will take some time.

  1. Lines 254, 520, 646, etc. nude mice, capital letters are not necessary.

Authors: Corrected. Please, see lines 256, 263, 537, and 664.

We hope that this version of the manuscript will be acceptable for publication.

Thank you very much!

Reviewer 3 Report

General comments:

The authors reported the design and synthesis of 11 novel amide derivatives of soloxolone methyl (SM) and identified the compound 12 exhibiting anti-glioblastoma effects with the in vitro and in vivo evidences.

Major comments:

  1. Table 1 & Table 2: Please provide the cell viability assay and drug time at the footnote as well as the repeat number (n = ?).
  2. What is the characters of glioblastoma cells such as U87 and U118? Please provide brief information of their basic information.
  3. Table 1: Compound 12 show similar IC50 values between cancer and normal cells (fibroblast). Please provide some discussions for it. What is the tissue source of the fibroblast? It is possible that these normal cell line and tumor cell lines were derived from different tissues and made it hard to compare?

Minor comments:

  1. p value in each figure legend should be typed in italic font.

Author Response

Dear Reviewer #3,

We are very grateful to you for the valuable comments and suggestions that helped us to improve the manuscript. We revised the manuscript according to your comments and, please, let us respond to your remarks.

  1. Table 1 & Table 2: Please provide the cell viability assay and drug time at the footnote as well as the repeat number (n = ?).

Authors: Corrected. The information about used cell viability assay, incubation time and the number of experimental repeats was added to the footnotes of Table 1 (please, see lines 166-167) and Table 2 (please, see lines 317-318).

  1. What is the characters of glioblastoma cells such as U87 and U118? Please provide brief information of their basic information.

Authors: Corrected. Thank you for your valuable comment. Indeed, it is our omission to explore U87 and U118 glioblastoma cells and not to give brief information about them. The description of key characteristics of U87 and U118 cells was added in Section 2.2.4 (please, see p. 9, lines 311-315).

  1. Table 1: Compound 12 show similar IC50 values between cancer and normal cells (fibroblast). Please provide some discussions for it. What is the tissue source of the fibroblast? It is possible that these normal cell line and tumor cell lines were derived from different tissues and made it hard to compare?

Authors: Yes, you are absolutely right. In our cytotoxic screening, we used tumor cells sourced from the tissues of the cervix of the uterus, duodenum, skin and brain as well as non-transformed foreskin fibroblasts. Thus, observed similar cytotoxic rates of soloxolone amides in malignant and normal cells indeed can be explained by the different tissue origin of the latter, which indicates the need of more careful selection of non-malignant control cells for cytotoxic profiling. In our opinion, to more comprehensively evaluate the tumor selectivity of the compounds, in vivo studies in tumor-bearing mice are required. Given the fact that 12 effectively suppressed growth of U87 xenograft in nude mice and did not induce general toxicity, this compound is indeed characterized by promising selectivity for glioblastoma cells. The text describing this information was introduced in the Section 2.2.1 (please, see p. 6, lines 197-200). Moreover, we indicated the tissue source of hFF3 fibroblasts in the Section 2.2.1 (Cytotoxic screening) and in Material and methods (please, see lines 146 and 950).

We hope that corrected version of the manuscript will be acceptable for publication in Pharmaceuticals.